# Automated and optimally FRET-assisted structural modeling

Mykola Dimura [1,2], Thomas-Otavio Peulen [1], Hugo Sanabria [1,3], Dmitro Rodnin[1], Katherina Hemmen [1], Christian A. Hanke [1], Claus A. M. Seidel [1✉] & Holger Gohlke [2,4✉]

FRET experiments can provide state-specific structural information of complex dynamic biomolecular assemblies. However, to overcome the sparsity of FRET experiments, they need to be combined with computer simulations. We introduce a program suite with (*i*) an automated design tool for FRET experiments, which determines how many and which FRET pairs should be used to minimize the uncertainty and maximize the accuracy of an integrative structure, (*ii*) an efficient approach for FRET-assisted coarse-grained structural modeling, and all-atom molecular dynamics simulations-based refinement, and (*iii*) a quantitative quality estimate for judging the accuracy of FRET-derived structures as opposed to precision. We benchmark our tools against simulated and experimental data of proteins with multiple conformational states and demonstrate an accuracy of ~3 Å $RMSD_{C\alpha}$ against X-ray structures for sets of 15 to 23 FRET pairs. Free and open-source software for the introduced workflow is available at https://github.com/Fluorescence-Tools. A web server for FRET-assisted structural modeling of proteins is available at http://nmsim.de.

[1] Chair for Molecular Physical Chemistry, Heinrich Heine University Düsseldorf, 40225 Düsseldorf, Germany. [2] Institute for Pharmaceutical and Medicinal Chemistry, Heinrich Heine University Düsseldorf, 40225 Düsseldorf, Germany. [3] Department of Physics and Astronomy, Clemson University, Clemson, SC, USA. [4] John von Neumann Institute for Computing (NIC), Jülich Supercomputing Centre (JSC), and Institute of Biological Information Processing (IBI-7: Structural Biochemistry), Forschungszentrum Jülich GmbH, 52425 Jülich, Germany. ✉email: cseidel@hhu.de; gohlke@uni-duesseldorf.de

Structures of biomacromolecules and their complexes are essential to understand the underlying molecular mechanisms of the biological processes. Being of key importance for rational drug design, biological and medical developments, experimental three-dimensional structures determined by X-ray crystallography, cryo-electron microscopy, and NMR are deposited[1,2]. If biomolecular systems are complex, information from multiple experimental and computational methods is combined by integrative modeling (IM) for generating integrative structure models. In this context, FRET experiments with quantitative analysis are increasingly used to provide dynamic information on the studied system and to determine integrative structures[3]. A prototype archive called PDB-Dev (pdb-dev. wwpdb.org)[4] has been established to collect such integrative structures.

For certain classes of systems, including multidomain proteins, biomacromolecular complexes, dynamic systems with unstructured regions, and systems with lowly populated conformational states, experimental structure determination is challenging. For such systems, contemporary computational structure prediction tools[5–9] often yield several competing models, which may contain different domain folds and supertertiary structures, particularly if template structures of homologous proteins are incomplete, as is frequently the case for multidomain proteins[10]. Quantitative FRET experiments on the single-molecule and ensemble level became especially popular[11–13], because they can alleviate these difficulties by state-specific structural information on complex constructs, even for dynamic systems with short-lived states in the microsecond time scale[14–18]. To overcome the issue that FRET data is too sparse to cover all structural details[11,12], FRET experiments need to be complemented with computer simulations.

Integrative modeling transforms experimental uncertainties to structural uncertainties. Here, we use the term experimental uncertainty for the statistical dispersion of measured values, including random and systematic uncertainties. In a single-molecule FRET experiment, these contributions correspond to the photon noise (random) and the intensity calibration of the detection channels (systematic). We characterize the conformational variability of FRET-assisted structures by the term uncertainty of the model. We use the term expected uncertainty of the model for an uncertainty estimated for a structural model based on the number and average quality of used measurements and the properties of the underlying computational method. We use the terms precision for the variability between repeated measurements, and accuracy for the closeness of agreement between a test result and the accepted reference value, e.g., RMSD of a FRET-assisted model with respect to the reference crystal structure.

Despite the value of FRET spectroscopy for restrained structural modeling, a number of issues challenge its practical application. In this work, we focus on issues that can be alleviated by improved algorithmic experiment design and modeling. One issue is that the planning of FRET experiments, i.e., determining informative labeling sites, remains to the intuition of the user. Moreover, it is unclear how many FRET pairs are needed to achieve the desired uncertainty of the model, and experiments can be designed suboptimal. Thus, a generally available automated design of informative FRET experiments, previously suggested[16] and described here, will minimize the total experimental effort by maximizing the information content per measurement.

Another issue is that studied biomolecules are usually labeled with flexibly coupled dyes in order for the dyes to change their orientation isotropically within the donor lifetime. Yet, this complicates the data analysis and the modeling steps, as the positional distribution of the dyes needs to be considered to correctly describe the experimental FRET observables[12].

However, it is difficult to recover this distribution at a high level of detail by traditional all-atom molecular dynamics (MD) simulations using explicit labels[19], because sampling of the dye linker configurations requires simulation times in the order of microseconds and longer. Moreover, it was shown recently that labeling positions that exhibit different anisotropy could not be treated consistently by a dye model that does not correct for the dye mobility[13]. Therefore, new dye models must be developed, which will account for the variable experimentally determined dye mobility[16,20,21].

A further challenge is to establish FRET-guided computer simulations to refine structures and model conformational transitions. The implementation of FRET restraints for flexibly tethered dyes is difficult since the long dye linkers act as soft entropic springs that can absorb the strain of the restraints. Thus, when a FRET-restraint is applied, it is reflected immediately by a corresponding displacement of the soft dye linker, but barely propagates to the protein backbone. To achieve controlled force transmission, we use an implicit dye model with a pseudoatom at the mean position of the dye distribution. In FRET-guided simulations, the mean position is restrained with respect to the adjacent protein backbone, and FRET-derived forces are applied between the mean positions. We refer to this procedure as a restrained mean position (RMP) approach.

Finally, statistically valid procedures for the quality assessment of FRET-restrained integrative structure models must be established. So far, the problem of a quantitative accuracy assessment, as opposed to precision, remains largely unaddressed. Such an accuracy estimate is essential for quality control of integrative/hybrid structure models, including those deposited to PDB-Dev[4]. In recent publications of FRET-assisted structural modeling[12,13,22], the FRET restraints were minimized during the structure generation to obtain structures with minimal reduced chi-square value $\chi_r^2$, see Eq. (7). However, absolute $\chi_r^2$ values were to some extent arbitrary due to the inaccuracy of the experimental uncertainty estimates and an ill-defined number of effective fitting parameters used to derive the structure model. Consequently, the overall quality of the structure model with respect to the FRET restraints and the model's statistical significance remained unclear. Recent advances in the standardization of FRET experiments in a multi-laboratory benchmark study[23] greatly improved the experimental procedures and optimized error estimation. Unfortunately, determining the effective number of parameters of a model used to determine a structure is very challenging, and no general and reliable methods are known to alleviate this issue directly[24,25]. Therefore, we suggest a workaround and define a reliable quality parameter and accuracy estimate for FRET-assisted structure models based on the cross-validation.

Altogether, we introduce a freely accessible software suite composed of a set of programs that address all issues mentioned above. The program suite FRET positioning system (FPS 2.0) introduces a function for experiment planning with automated FRET-pair selection and provides improved dye models, in addition to the previously available calculation of FRET observables, FRET-restrained rigid-body docking, and screening (https://github.com/Fluorescence-Tools/Olga). Aiming at the FRET-assisted refinement of conformers or docked substructures, we developed two approaches that explicitly consider positional dye distributions and apply FRET restraints to the biomolecular system via the RMP approach: the command-line tool FRETrest for FRET-restrained molecular dynamics simulations (https://github.com/Fluorescence-Tools/FRETrest); and a FRET-guided coarse-grained simulation approach based on the NMSim geometric conformational sampling software (http://nmsim.de)[26]. Finally, we provide a library for coarse-

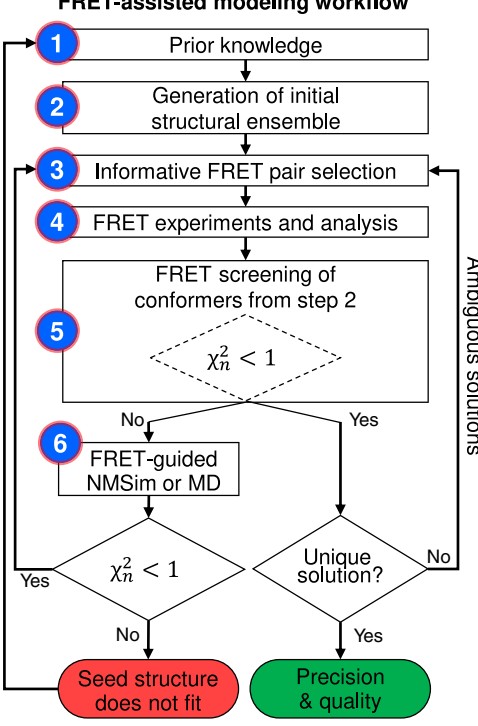

**FRET-assisted modeling workflow**

**Fig. 1 Iterative workflow for automated and optimally FRET-assisted structural modeling.** The workflow covers cases with (left branch) and without FRET-guided conformer optimization (right branch) and using a $\chi_n^2$ as a criterion for branching ("Methods", Eq. (8)).

grained simulations of fluorescent dyes called LabelLib (https://github.com/Fluorescence-Tools/LabelLib), which can also be incorporated in third-party integrative modeling tools. We demonstrate the usefulness of this suite and validate it on flexible proteins using interdye distances calculated from simulated typical single-molecule FRET data of five proteins and experimental FRET data of lysozyme of the bacteriophage T4 (T4L)[27].

## Results

**Workflow for automated FRET-assisted modeling**. We devised an iterative six step workflow for FRET-assisted modeling and developed a software suite for all steps (Fig. 1): (step 1) collection of prior knowledge, (step 2) generation of an initial structural ensemble, (step 3) automated selection of the most informative FRET pairs, (step 4) acquisition and analysis of experimental data, and (step 5) FRET screening based on statistical quality assessment using the $\chi_n^2$ criterion, see Eq. (8). Depending on the results of step 5, the conformers can be optimized by FRET-guided simulations using FRET-guided NMSim and FRETrest (step 6). The implemented workflow is exemplified for the *E. coli* protein YaaA using simulated FRET data (Fig. 2).

**Workflow without FRET-guided conformer optimization**. In step 1, prior structural knowledge is gathered, e.g., from structures in the PDB of other states of a given target, comparative models, or structure models built with other computational structure prediction tools[5–9] (Fig. 2a). For YaaA, prior knowledge was taken from the computational structure predictions submitted to the CASP11 experiment (T806)[28] (see the section "Proteins used in the benchmark"). The diversity of the YaaA ensemble with respect to the secondary structure, folds, and conformations is displayed in Fig. 2a. For all other targets of this benchmark, the seed structures and the "true" target structures

were taken from the PDB (Table 1). Note that the seed and target structures differ in all cases in their conformational state (see the section "Proteins used in the benchmark" and Table 1). In step 2, the structural ensemble of the seed structures is expanded by conformational sampling to have a more continuous and complete coverage of the conformational space (Fig. 2b). This is achieved by multiple unrestrained simulations using the structures obtained in the first step as seeds. Here, we used NMSim to expand the initial ensemble. NMSim performs normal mode-based geometric simulations for multiscale modeling of protein conformational changes[26]. In step 3, we realize our recent suggestions for an automated design of FRET experiments[16]. To optimize FRET experiments, our program suite FPS 2.0 uses a feature selection algorithm to determine automatically the smallest set of most informative FRET pairs (Supplementary Fig. 1 and Supplementary Table 1), which minimizes the expected uncertainty of the model for a given initial structural ensemble (see the section "Workflow: selection of a set of optimal FRET pairs"). By this procedure, we answer the questions of how many and which FRET pairs should be measured to resolve uncertainties in a structure of a protein or a complex. The selection algorithm searches for the FRET pairs that are best aligned to the directions of potential protein motion, that are least redundant with respect to each other, and that are expected to produce distances close to the Förster radius, where the sensitivity of the measurements is the highest.

Additionally, FPS 2.0 can consider user-specified labeling site accessibilities, their chemical nature, and influence on the function and stability as determined from mutation analysis or sequence coevolution data (see the section "Workflow: selection of a set of optimal FRET pairs"). The circular plot in Fig. 2c depicts the location of the FRET pairs suggested for protein YaaA to interrogate the different folds and conformations of the initial ensemble. In step 4, FRET data including estimates of experimental uncertainty are acquired. To maximize the significance of this benchmark, the uncertainty estimates of the simulated datasets were matched to experimental data measured for T4L[23] (Supplementary Table 1).

In step 5, structures in agreement with the FRET observables are identified. Here, we use FPS 2.0 to score the conformers in a structural ensemble by their agreement with the FRET observables. This agreement is often quantified using the reduced chi-squared $\chi_r^2$ (Eq. (7)). However, for structure models with different numbers of degrees of freedom, a constant confidence level corresponds to different values of $\chi_r^2$, making comparisons inconvenient (Supplementary Fig. 2). To compare models with different numbers of degrees of freedom, we introduce $\chi_n^2 = \chi^2/\chi_{68\%}^2$ (Eq. (8)) as a quantitative and reliable accuracy estimate. This way, models of different complexity, e.g., generated by rigid-body docking and molecular dynamics simulations, can be compared on one graph, as for any $\chi_n^2 = 1$ the confidence level is 68%. The absolute measurement of quality $\chi_n^2$ relies on accurate error estimates and FRET measurements not previously used in model optimization. Therefore, $\chi_n^2$ corresponds to cross-validation of the structure model and is similar in spirit to the $R_{\text{free}}$ known from X-ray crystallography[29].

To calculate $\chi_n^2$, FPS 2.0 introduces a function that automatically estimates the number of relevant degrees of freedom for FRET-assisted models. The tool can be applied to an arbitrary ensemble of structures and provides a convenient interface for third-party structural modeling tools. Using the quality parameter $\chi_n^2$, the FRET data allow us to extract a set of FRET-consistent models by identifying structures with $\chi_n^2$ values <1.0 (Supplementary Fig. 3). A higher number of measured informative FRET pairs decreases the model uncertainty, since less diverse structures

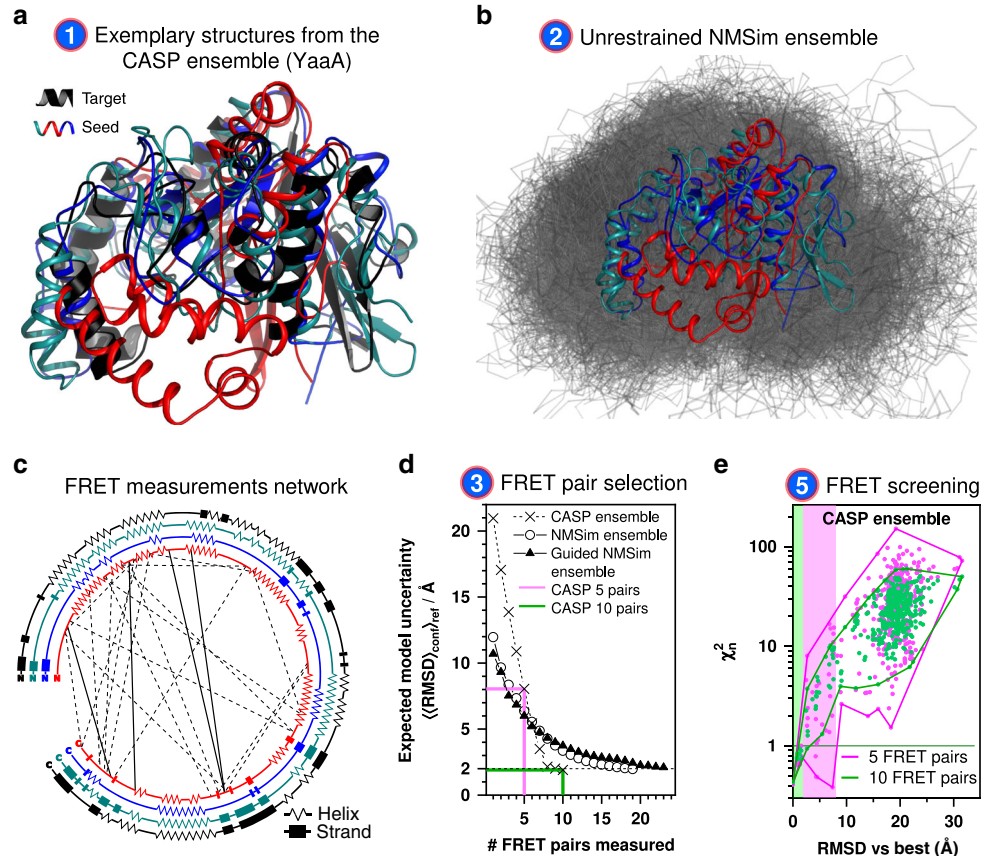

**Fig. 2 Automated FRET-assisted structure prediction demonstrated on the *E. coli* protein YaaA. a** Collection of structures used as prior knowledge. The CASP predictions serving as seed structures (red, cyan, blue), and the target crystal structure of YaaA (black, PDB ID: 5CAJ) are shown as cartoon representations. The conformers differ in their secondary structure and folds (see also **c**). For clarity, only three out of the ten used seed structures are shown. **b** Generation of the initial structural ensemble (gray) by NMSim without FRET information, using the CASP predictions (red, cyan, blue) as seeds. **c** Network of FRET pairs used for FRET-guided NMSim (dashed lines) and screening (dashed and solid lines). Secondary structure elements of the three shown seed structures (red, cyan, blue) and the target (black) are represented by zigzags (α-, $3_{10}$-, or π-helices), rectangles (β-sheets), and lines (loops). **d** Expected uncertainty of the resulting structure, depending on the number of FRET measurements used for modeling. For sparse conformational ensembles as the CASP ensemble (crosses), the decay is steeper compared to the dense ensembles generated by NMSim (circles). **e** Impact of the number of FRET restraints on the uncertainty of the selected ensemble. The $\chi_n^2$ values and RMSD vs. the structure in best agreement with the simulated or experimental information is shown for the structural ensemble of the CASP targets. The set of the structures with a $\chi_n^2$ lower than 1.0 defines the uncertainty of the FRET-selected structure. The green and magenta shaded areas correspond to 10 and 5 FRET measurements, respectively. Source data are provided as a Source Data file.

### Table 1 Summary for the proteins used in this benchmark study[a].

| Protein name | PDB ID (seed/target) | #aa | RMSD$_{seed}$/Å | RMSD$_{best}$[$-\Delta$, $+\Delta$]/Å | #pairs (guiding + validation)[c] | Granularity #FRET pairs |
|---|---|---|---|---|---|---|
| *E. coli* YaaA protein | [b]/5caj | 256 | 4.7–14.6 | 2.4 [$-0.2$, $+0.1$] | 19 + 4 | 1:11.1 |
| Adenylate kinase | 4ake/1ake | 214 | 7.2 | 2.3 [$-0.2$, $+0.9$] | 10 + 8 | 1:11.9 |
| LAO-binding protein | 2lao/1lst | 238 | 4.7 | 2.4 [$-0.6$, $+0.1$] | 12 + 3 | 1:15.9 |
| Calmodulin | 1cfd/1ckk | 148 | 9.8 | 2.4 [$-0.1$, $+0.7$] | 13 + 9 | 1:6.7 |
| Atlastin-1 | 4idn/3q5e | 409 | 18.7 | 2.5 [$-0.1$, $+0.5$] | 10 + 9 | 1:21.5 |
| T4 lysozyme (C2 → C1) | 3gun/172l | 162 | 4.0 | 2.8 [$-0.1$, $+0.5$] | 10 + 10 | 1:8.1 |
| T4 lysozyme (C1 → C2) | 172l/3gun | 162 | 4.0 | 2.5 [$-0.5$, $+1.0$] | 10 + 10 | 1:8.1 |

#aa stands for the number of amino acids of the protein used in the benchmark study. The root mean square deviation (RMSD) of the seed structure against the target structure is indicated as RMSD$_{seed}$. RMSD of the FRET-selected structures against the target structure is indicated as an accuracy measure for the obtained ensembles; RMSD$_{best}$ represents the deviation from the target structure for the model with the lowest $\chi_n^2$; $-\Delta$ and $+\Delta$ indicate the range of RMSD values for the FRET-selected structure models within the 1σ confidence interval. All RMSDs are calculated for C$_\alpha$ atoms only. For T4L (underlined) experimental FRET data were used; for other proteins, the FRET data were simulated. Granularity is defined as the ratio of the number of FRET pairs required to achieve a specified uncertainty of the model (guiding + validation) and the number of amino acids in the protein.
[a]The starting ensembles, FRET networks, and optimization cycles are summarized for all proteins in Supplementary Fig. 8.
[b]For *E. coli* YaaA protein, 10 seed structures were selected among the predictions submitted for the CASP11 experiment (target T806). This selection differs from the target crystal structure (RMSD of 4.7–14.6 Å) and represents different folds and secondary structures.
[c]The number of FRET measurements needed for reliable segregation of models is reported in the #pairs column. Initially predicted FRET pairs are used for guiding, while an extended set of FRET pairs is used for cross-validation.

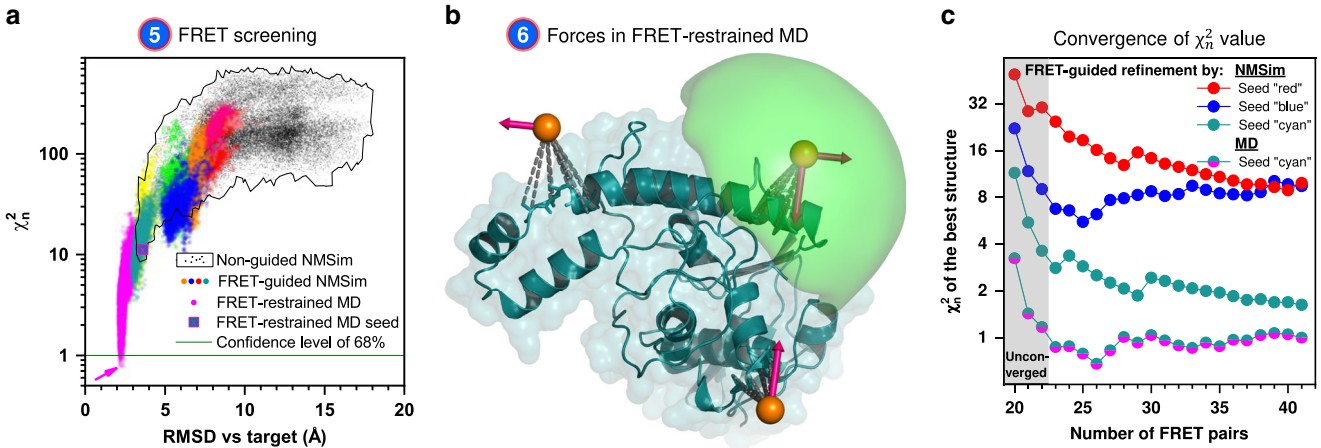

**Fig. 3 FRET-guided modeling of the *E. coli* protein YaaA. a** FRET $\chi_n^2$ values and RMSD against the crystal structure (target) of different conformations (points). The black points represent models obtained by unrestrained NMSim sampling starting from homology models. Colored points represent FRET-guided NMSim simulations. The magenta points represent FRET-restrained MD simulations. Guided simulations starting from different homology models are shown in different colors. The magenta arrow points to the structure with the lowest $\chi_n^2$. **b** Attachment (dashed gray) of pseudoatoms (orange spheres) and application of FRET restraints (pink arrows) in FRET-restrained MD simulations. The accessible volume of a fluorophore is shown as green surface. **c** $\chi_n^2$ of the FRET-guided models depending on the number of cross-validation FRET pairs, number of fit parameters is constant ($N_{\text{fit.param.}} = 19$). Each curve represents a best conformer generated by FRET-guided NMSim or FRET-restrained MD simulations using different seed structures. $\chi_n^2$ starts to converge with ~23 selected FRET pairs (four cross-validation FRET pairs), where the curves reach a plateau. The line colors correspond to the color of the structures in Fig. 2a. Source data are provided as a Source Data file.

are identified from the ensemble within the confidence level of 68% corresponding to the $\chi_n^2 < 1.0$ (Fig. 2e). This decrease of the model uncertainty agrees with the predictions from the pair selection algorithm (Fig. 2d). Our estimate of the model uncertainty is only valid for a concrete initial ensemble, i.e., for the sparser and smaller initial ensemble from CASP11 (Fig. 2a) fewer FRET measurements are needed to achieve the desired expected uncertainty than for the larger and more detailed ensemble expanded by NMSim (Fig. 2b). A larger initial ensemble corresponds to a larger space of candidate conformations and, generally, requires more FRET measurements to achieve the target model uncertainty. If the diversity within the FRET-selected ensemble is sufficiently low (e.g., root-mean-square deviation of any structure pair $\text{RMSD}_{ij} < 3$ Å), the workflow is considered converged (right side of the workflow in Fig. 1). The heterogeneity of the FRET-selected ensemble represents the uncertainty of the obtained model (Fig. 2e).

**FRET-guided optimization of conformers**. In step 6 (left branch of the workflow in Fig. 1), we optimize the structures if no conformer with good agreement with the FRET data ($\chi_n^2 < 1$) was found in the initial ensemble (Fig. 3a, black points). We established two structural sampling schemes for FRET-guided structural sampling: FRET-guided normal mode-based geometric simulations (NMSim)[26] (Supplementary Fig. 4), employing a Metropolis–Hastings Monte Carlo algorithm, and FRET-restrained molecular dynamics simulations via the tool FRETrest (Fig. 3b and Supplementary Fig. 5). In both schemes, we implemented an implicit coarse-grained dye representation to model experiment-based interdye distance restraints, rather than using inaccurate atom–atom distance restraints. In our RMP approach, FRET restraints are applied to the mean position of the dye distribution and not to the explicit dye. The pseudoatom that represents the mean dye position is strongly restrained with respect to the protein backbone by pseudobonds and cannot diffuse, such that the force applied to it is transmitted directly to the backbone. The additional FRET information allows us to explore areas of the phase space that are difficult to access for

purely computational multiscale simulations, such that novel and experimentally relevant (super-)tertiary structures can be resolved (colored populations in Fig. 3a). FRET-guided refinement of different seed structures yields distinct $\chi_n^2$ levels for the final structure models (Fig. 3a, c) with more accurate folds indicated by lower $\chi_n^2$ values. This allows us to detect incorrect folds of seed conformers (Fig. 2a, c; conformers depicted in red and blue) that cannot be easily corrected. Note that for this refinement, only four additional FRET pairs are needed for the conformer depicted in cyan (NMSim) and cyan–magenta (MD) for reaching a converged $\chi_n^2$ (gray box, Fig. 3c). In turn, if the initial ensemble does not contain structures with correct secondary structure or fold, FRET-guided optimization procedures, as used here, will not be able to find the right structure. Yet, this situation is clearly detected by $\chi_n^2 \gg 1$, which indicates that a better initial ensemble is required.

**Benchmarking the methodology**. The workflow was benchmarked on simulated and experimental data. For that, we used an exemplary set of six proteins (YaaA of *E. coli*, Adenylate kinase, LAO-binding protein, Calmodulin, Atlastin-1, and T4 lysozyme), listed in Table 1, that are diverse in their structures, sizes (148–409 amino acids), and types of internal interconversion motions (hinge-bending, shear, and twist), and mode of interaction (induced fit or conformational selection[30,31], Supplementary Note 1). Some of these proteins have been used previously to investigate conformational sampling techniques[32–34]. For each protein, detailed structural information on at least one conformation is available in the PDB. Here, this conformation is used as a 'true' reference structure for accuracy estimation. For five proteins, realistic FRET data were simulated[16] in that average interdye distances were calculated from accessible volume (AV) simulations using the crystal structure of the target state. We compute the error of the average interdye distance by propagating the absolute error of the FRET efficiency $\Delta E = 0.06$. This error value is typical for single-molecule FRET measurements as determined in a community-wide benchmark study[23] (Supplementary Table 1 and Supplementary Note 2). Additionally, for T4

lysozyme (T4L), a comprehensive experimental data set was acquired in solution, which allowed us to resolve two short-lived (4 μs) conformers referred to as C1 and C2[27], which were similar to conformers observed previously by X-ray crystallography. Using simulated and experimental datasets for target structures in Table 1, we applied our FRET-guided structural modeling procedure in order to arrive at a target structure model, starting from the seed conformer corresponding to the other state. In this benchmark study, we obtained state-specific ensembles of structure models with an uncertainty of 2–3.5 Å and accuracy against the target structure between 2 and 3 Å (Fig. 4, Table 1, and Supplementary Figs. 6, 7) for as few as 15 to 23 FRET measurements. The parsimony in FRET pairs is attributed to the method for the automatic determination of a set of optimal FRET pairs (Supplementary Fig. 1). The accuracy of obtained models and the number of necessary FRET pairs depends on the structural diversity and accuracy of the initial ensemble (Fig. 2d). To achieve the same uncertainty of the model, different numbers of FRET pairs per numbers of residues (Table 1) were necessary, since the size of structural elements varies greatly between proteins and models. We designed initial ensembles in this benchmark to mimic the inaccuracy and conformational uncertainty expected from current computational modeling tools, like those demonstrated in CASP[35]. These results illustrate that the predictive power and reliability of $\chi_n^2$ (Supplementary Fig. 7) yields target structures with an observed structural heterogeneity for protein backbone conformations at room temperature as found in all-atom MD simulations and NMR experiments[36]. The resolution and information content of experimental FRET studies are sufficient to distinguish between the known conformers C1 and C2 (Supplementary Fig. 6), which differ by 4 Å RMSD.

**Assessment of the local model accuracy.** The resolution of FRET data, quantified by the uncertainty of interdye distances, can be as high as close to 1 Å. Of utmost importance is to describe the dye motions accurately to achieve a well-defined dye localization to draw conclusions on the underlying structural model. Still, due to the sparse nature of FRET data, a direct conversion of experimental uncertainties to model uncertainties is not trivial. By contrast, resolution in X-ray crystallography relates to the experimental data itself, i.e., resolution of the electron density map, which is closely related to the uncertainty of the X-ray-based structural model. Therefore, to assess local model accuracy, we provide a toolkit to propagate the uncertainties in FRET scoring. The number of FRET measurements, required to achieve a specified uncertainty of a model, depends on the complexity of this model. For example, if we describe the conformation of a protein backbone only by the phi and psi dihedral angles, then the number of fit parameters in such a model is $2 \times (N_{AA} - 1)$[37]. At the same time, the number of measured FRET distances is typically more than an order of magnitude lower. This disparity is referred to as the sparsity of the FRET data. However, modern computational methods restrict the number of fit parameters in the protein model by applying stereochemical constraints, constraints derived from homology modeling, and coevolution-based contact predictions, which reduces the granularity required for a protein model. We quantify the granularity of a model by the ratio of the number of FRET pairs required to achieve a specified uncertainty of the model to the number of amino acids in the protein (Table 1). In this benchmark, the granularity varies from 1 FRET pair per 6.7 amino acids for the locally flexible calmodulin to one pair per 21.5 amino acids for atlastin-1, reflecting that atlastin-1 consists of fewer, larger, and less flexible subdomains.

Since individual FRET measurements provide information specific to the pair of residues being labeled, and typically <30% of residues are labeled, the accuracy can vary for different parts of the model. We illustrate the variations of local accuracy for the protein YaaA (Fig. 5) by calculating the average (Fig. 5a) and residue-specific (Fig. 5b) local distance difference test (lDDT) superposition-free similarity scores[38] against the crystal structure. The lDDT score of an atom represents the fraction of conserved distances to other atoms within the inclusion radius of 15 Å and, thus, captures secondary structure differences better than RMSD and does not suffer from alignment-sensitivity (see Supplementary Note 3). Both lDDT (Fig. 5a) and RMSD (Fig. 3) correlate very similarly with $\chi_n^2$.

To demonstrate the effect of the sparsity of FRET data, we compared global lDDT scores, calculated for all amino acids in the protein, to the average lDDT scores of amino acids labeled with donor and acceptor, respectively ($lDDT_D$, $lDDT_A$) (Fig. 5a). For reference, we also show the spread of lDDT scores caused by thermal fluctuations, as observed for 500 ns of unrestrained MD simulations starting from the crystal structure of YaaA (Fig. 5a, gray points). The FRET-selected conformers have global lDDT scores of 0.7 ± 0.02 (Fig. 5b, magenta line), whereas structures in the MD ensemble have global lDDT scores of 0.9 ± 0.02 (Fig. 5b, cyan line). Hence, the accuracy of FRET-selected conformers is 0.2 lDDT points lower than what is observed purely due to thermal fluctuations. At least half of this difference can be attributed to the sparsity of labeling, since lDDT scores of FRET-labeled residues is 0.77 ± 0.02. The least accurately resolved residues are positioned in the hydrophobic core of the protein (residues 109–121) and near it (residues 73–81), as evident from the solvent-accessible surface area (Fig. 5b, green dotted line). FRET-guided modeling cannot correct these inaccuracies since labeling of buried residues is impossible and refolding of the secondary structure requires extensive all-atom MD simulations. Lower lDDT values are also noticeable for loop regions (residues 17–25, 138–141, 160–162, 204–206) that are more flexible, as indicated by the larger spread of the lDDT values in the MD simulations (Fig. 5b, cyan-shaded area).

## Discussion

Using simulated and experimental data, we demonstrate that accurate, efficient, and largely automated protein structure determination is possible based on optimally designed FRET experiments and structural modeling at multiple scales. Our approach is based on accurate state-specific interdye distance measurements, typically obtained via intensity-based single-molecule fluorescence techniques such as dynamic Photon Distribution Analysis (PDA)[39], hidden Markov Modeling[40], or analysis of fluorescence decays obtained from bulk measurements[41]. Crucially, these methods utilize the Poisson statistics of single-photon counting, so that quantitative error estimates for the estimated distances are obtained, which enable the quantitative quality assessment of integrative models. One important feature of single-molecule fluorescence techniques is their high time resolution for characterizing short-lived conformations: (i) intensity-based parameters allow one to quantitatively separate states that exchange at timescales of 100 microseconds or slower; and (ii) analysis of fluorescence decays recovers state-specific distance distributions even for state transitions with submicrosecond timescales. Experimentally, the distance is recovered together with the corresponding fractions, which helps to attribute the distances to individual states with different populations (fractions).

We extended our FPS toolkit to a FRET suite to cover the whole procedure from experiment design to the validation of the

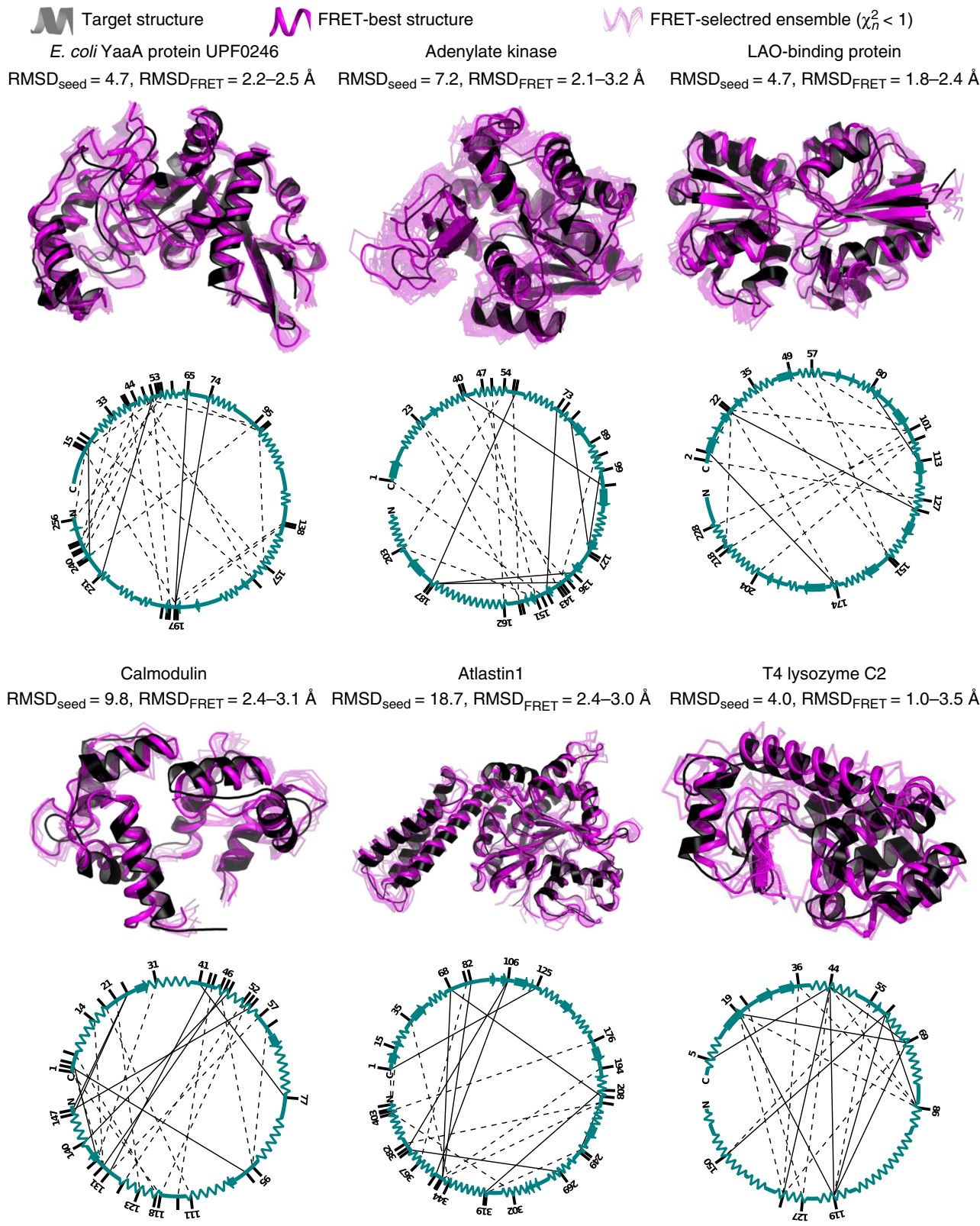

**Fig. 4 Accuracy and uncertainty of FRET-assisted structure models.** Structures obtained by FRET-assisted modeling (magenta) and target X-ray structures (black) are shown for each of the benchmarked proteins. FRET-selected structures are depicted in transparent magenta as a measure for uncertainty; a confidence level of 68% is assumed. RMSD of $C_\alpha$ atoms of the seed conformer ($RMSD_{seed}$) and FRET-selected conformers ($RMSD_{FRET}$) against the target crystal structure is shown below the protein name. Networks of FRET pairs and secondary structures of corresponding seed conformers (cyan) are shown below the selected ensembles.

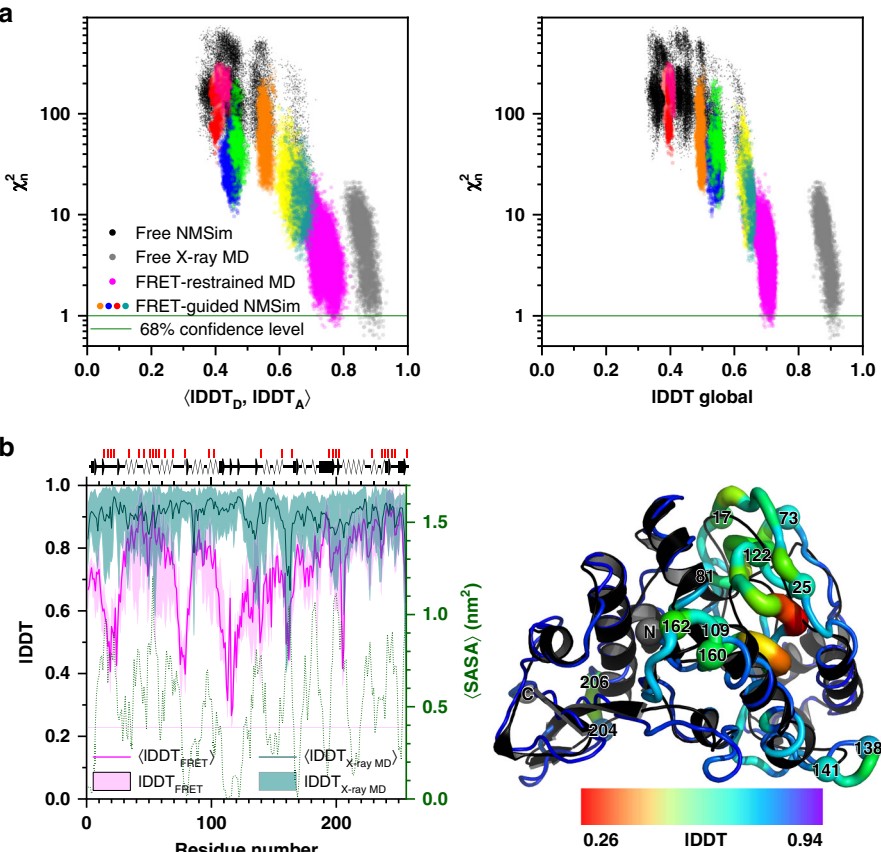

**Fig. 5 Local accuracy and uncertainty of FRET-assisted structure models of the E. coli protein YaaA. a** The left panel displays FRET $\chi_n^2$ values and local distance difference test (lDDT) scores against the crystal structure (target) for different conformations (points). $\langle \text{lDDT}_D, \text{lDDT}_A \rangle$ stands for the average over lDDT scores for residues that were in silico labeled by donor (D) and acceptor (A) fluorophores. Black points stand for unrestrained NMSim sampling starting from homology models. Colored points represent FRET-guided NMSim simulations. Magenta points represent FRET-restrained MD simulations. Guided simulations stemming from different homology models are shown in different colors. Gray points represent unrestrained MD simulations, started from the crystal structure of the YaaA protein, for reference. In the right panel, the x axis shows global lDDT scores: average number of conserved distances in the structure over four tolerance thresholds: 0.5, 1, 2, and 4 Å[38]. The lDDT scores were computed using only distances between α carbon atoms. **b** The magenta line represents ensemble-average lDDT score for FRET-selected conformers ($\langle \text{lDDT}_{FRET} \rangle$). The shaded magenta area represents the range of lDDT scores in the FRET-selected ensemble (lDDT$_{FRET}$). The cyan line represents ensemble-average lDDT for conformers from 500 ns of unrestrained MD simulations started from the crystal structure ($\langle \text{lDDT}_{Xray\ MD} \rangle$), the shaded area indicates the range of lDDT scores in this simulation (lDDT$_{X-ray\ MD}$). The green dotted line shows the solvent-accessible surface area averaged over 500 ns of unrestrained MD simulations and smoothed over the window of five residues. Secondary structure elements of the crystal structure are shown above the graph. Red vertical lines on top of the secondary structure information indicate the labeling positions. The putty plot shows the FRET-selected conformer; the color and thickness of the tube represent the residue-wise lDDT against the crystal structure (black cartoon), red thick regions correspond to less accurately predicted residues. Data are provided in the Source Data file.

obtained integrative structure model (Fig. 1). We added three features crucial for the generation of FRET-derived structural models of defined quality: (i) planning of experiments, (ii) a methodology for quality assessment of FRET-assisted structural models, (iii) model refinement by FRET-guided NMSim and MD simulations.

(Feature i) Planning of FRET experiments and estimating the number of the degrees of freedom for a given initial ensemble. We provide experimentalists with a method and the software to determine the most informative FRET pairs and estimate the minimum number of measurements in advance, given the desired conformational uncertainty and the initial ensemble of models. For this, we developed a tool to estimate the number of relevant free parameters in FRET-assisted structure models. This number is evident from the dependency of the expected uncertainty on the number of measurements with distinct informative FRET pairs for a given initial ensemble (Fig. 2d). The number of fit parameters reflects the flexibility of the structural model and can

be thought of as a quantity proportional to the number of sub-domains or segments that move with respect to each other. A common approach for localizing rigid bodies is trilateration, where three or more distances are measured from each label. This approach is especially useful if a good initial model is not available. However, it is not necessarily the most efficient approach in the case of FRET-assisted hybrid modeling. Our pair selection algorithm picks the FRET pairs such that all these segments are covered, aligning the pairs along the directions of motion in the model and placing the pairs closer to the center of the dynamic range, so that the experimental uncertainty is minimized.

(Feature ii) Assessing the quality of FRET-assisted structure models. Proteins typically consist of hundreds of amino acids, and therefore, detailed structure models can require hundreds and thousands of free parameters to describe a conformation. Since FRET data are sparse (typically less than hundred FRET pairs), overfitting of the structure model to the FRET data is likely, resulting in multiple solutions with $\chi^2/N_{measurements} \ll 1$

and undefined $\chi_r^2$, since $N_{dof} = N_{measurements} - N_{fit.param.} \leq 0$ (Eq. (3)). This could result in broad FRET-selected structure ensembles and inaccurate models of uncertain quality (false positives). A comprehensive integrative structure model must include a quality estimate, but, as outlined before, an accurate estimate for the number of fit parameters in FRET-guided structural models is frequently difficult to obtain. However, if the number of fit parameters in the FRET-assisted structural model were known, $\chi_r^2$ or $\chi_n^2$ could be used to evaluate its quality. Cross-validation can be used as a workaround for this issue, i.e., $N_{fit.param.}$ stays fixed, while $N_{measurements}$ increases. However, if the number of independent observables is too small, the $\chi_n^2$ criterion is not fully reliable (see the gray area in Fig. 3c). In this case, additional measurements should be added until $\chi_n^2$ reaches a plateau. An estimate on the minimum number of these additional measurements is provided by the analysis of the FRET-guided ensemble produced at the previous iteration of the workflow (Supplementary Fig. 7).

(Feature iii) Model refinement. In this study, FRET-guided simulations are set up such that the secondary structure and the fold of the optimized conformer are very stable. Under these conditions, cross-validation allows us to detect wrong folds, since $\chi_n^2$ does not reach 1 (Fig. 3c). For example, in the case of YaaA protein, 19 FRET pairs were employed in the first iteration of the workflow (Fig. 2d). All these pairs were used for FRET-guided simulations ($N_{dof} = 19 - 19 = 0$), so that for cross-validation in the second round the number of fit parameters in the structural model is 19. Thus, four additional independent measurements are used for the calculation of $\chi_n^2$ in the second round ($N_{dof} = 23 - 19 = 4$). $\chi_n^2$ then reaches a plateau (Fig. 3c). In our benchmark study of the other proteins, we demonstrated that an additional 3–10 FRET pairs were necessary to provide independent (non-fitted) observables, such that $\chi_n^2$ value can be calculated and used to distinguish overfitted and inaccurate structures from the true positive solutions. In summary, cross-validation allows one to estimate the absolute quality of the structural model, as opposed to the uncertainty, which is typically estimated using support plane analysis, bootstrapping, or other similar statistical methods[12].

In this study, we describe several applications of this approach, where we build FRET-derived structural models. We use simulated experimental data to demonstrate that the orientation of subdomains can be recovered with an accuracy of 2–3 Å, using FRET-guided structural modeling with NMSim, given a conformation with correct secondary structure and fold in the initial structural ensemble (Table 1 and Fig. 4). Our approach to integrative modeling relies on the FRET data for information on the mutual orientation of subdomains and for validation of obtained structures, and uses computational modeling to fill in more detail, such as detailed secondary structure composition or side-chain orientations. Based on experimental data, we successfully recover conformations for two short-lived states of T4 lysozyme, which exchange at 4 μs[27]. This approach can also be used to provide structural information on transient states of biomolecules, which is often difficult to obtain by traditional methods, so that dynamic structural biology becomes feasible even for short-lived states[42]. We deposited the obtained structural and kinetic models to the archiving system for structure models obtained through integrative/hybrid methods, PDB-dev[43] (PDB-dev ID: PDBDEV_00000044). We also demonstrate the successful refinement of a structure model by FRET-restrained all-atom molecular dynamics simulations using the RMP approach.

The workflow can be applied to all biomacromolecules and biomacromolecular assemblies, including soluble and membrane proteins, as well as nucleic acids. The programs for structural modeling in the suite can be applied to structured biomolecules and to biomolecules with disordered regions. However, structural analysis of disordered regions is fundamentally challenging, yet several experimental approaches were proven fruitful, with FRET being among the most effective tools for mapping disordered systems[44–46]. Rapid fluctuations of disordered systems cause signal averaging, mandating non-trivial modeling, possibly using special force fields[47–49] or other approaches.

We demonstrated that FRET measurements detect changes in interdye distance caused by the packing of subdomains or their mutual reorientation (Fig. 5). Our approach to FRET-assisted modeling performs best when a structural model with a limited number of subdomains or segments is available, such that the total number of fit parameters in the structural model is less than the number of feasible FRET measurements. Secondary structure can be sensed by FRET indirectly if it affects the packing of subdomains. However, positions of fluorescent labels are almost unaffected by side-chain motions, or changes in the unlabeled interior of the protein, if they are not manifested by corresponding movements of surface residues. Complementarily, additional fluorescence information could be harvested for structural modeling from the quenching effects of buried fluorophores. For instance, FRET could be nicely combined with quenching studies by photoinduced electron transfer[41,50,51] to inform structural modeling on short distances.

In our view, the obtained results provide a major step ahead for quantitative FRET-assisted structural modeling. Altogether, these advances pave the way for convenient usage of FRET measurements in data-assisted computational modeling challenges within CASP and CAPRI[35]. The presented experiment design and modeling approach is independent of details on how structures are scored by experimental information. Thus, it is transferable to other techniques that inform on distances and use probes introduced at specific sites such as EPR[52], paramagnetic relaxation enhancement NMR[53], and vibrational spectroscopy[54] and will benefit from future improvements in the analysis of primary fluorescence data. Current FRET structures are based on experimentally derived distances, and corresponding uncertainty estimates, even though the primary data contains more information. However, due to the diverse types of fluorescence experiments, a scoring of structures directly by the primary experimental data is difficult and generally requires spectroscopic expertise. Currently, the FRET community (http://fret.community) starts initiatives to agree on data exchange formats, documentation of experiments, and analysis procedures. This will enable automated data-processing pipelines that can tightly couple primary data to structural modeling[41,50,51]. The complexity arising from different experiments could be mitigated by Bayesian hierarchical data-processing frameworks that abstract experimental data and propagate information and uncertainties to enable structural modeling at higher precision and accuracy.

## Methods

**Proteins used in the benchmark.** Six proteins were selected as targets to assess the performance of our FRET-assisted modeling approach: LAO-binding protein, adenylate kinase, calmodulin, atlastin-1, *E. coli* YaaA protein, and T4 lysozyme.

For the first four proteins[55–63], at least two crystal structures are known. One crystal structure was considered the target structure; the other structure(s) of a different conformational state(s) was(were) used as prior knowledge (Table 1). The *E. coli* YaaA protein was a target (T806, PDB ID 5caj) of the CASP11 experiment[28]. For YaaA, ten homology models of the CASP11 participants were used as prior information. The ten seed structures were selected from a pool of 639 complete protein models submitted to the CASP11 experiment by, first, removing structures that are similar to the target ($C_\alpha$ atom RMSD <4.6 Å). Next, the remaining 589 models were clustered using hierarchical agglomerative clustering[64] into 100 groups by their secondary structure. From the resulting 100 cluster representatives, 10 structures were manually selected, such that they represent different tertiary structures and different RMSDs ($C_\alpha$ atoms) with respect to the target (4.6 ≤ RMSD ≤ 14.6 Å). The structures of following CASP model IDs were used as the source of prior knowledge (seed conformers): Tc806TS041_1, Tc806TS065_1,

Tc806TS276_1, Tc806TS345_1, Tc806TS357_1, Tc806TS420_1, Tc806TS428_1, Tp806TS065_1, Ts806TS065_1, Ts806TS276_1.

**Theory: quality metric to evaluate the information content of sets of FRET pairs.** To assess how well a certain set of DA pairs helps to resolve a protein structure, we introduce the quality parameter $\langle\langle RMSD_{\#conf}\rangle_{\#ref}\rangle$, or short $\langle\langle RMSD\rangle\rangle$, as a measure to estimate for the expected uncertainty of the model (Supplementary Fig. 9).

Conceptually, to compute $\langle\langle RMSD\rangle\rangle$, we first take an arbitrary reference model from the pool of structure models and assume that it corresponds to the "true" structure of the molecule in the experiment. For this reference, a full set of FRET observables is computed. Next, the FRET observables are simulated for all conformers in the pool of structures and compared against the reference set of simulated observables. The simulated observables of the structures are compared to the reference using a $\chi^2$ test with corresponding $P$ values to yield a conformational uncertainty $\langle RMSD_{\#conf}\rangle$ for this reference. This procedure is repeated for all structures from the pool of structures, and the average over all $\langle RMSD_{\#conf}\rangle$ is calculated to yield $\langle\langle RMSD_{\#conf}\rangle_{\#ref}\rangle$ (Supplementary Fig. 9).

In practice, for a given set of $N$ structures, $\langle\langle RMSD\rangle\rangle$ is calculated in three steps (Supplementary Fig. 9): in the first step, pairwise RMSDs are computed for all combination of structures.

$$RMSD = RMSD_{conf,ref} = \sqrt{\frac{1}{N_{atoms}}\sum_{at=1}^{N_{atoms}}||\overrightarrow{r_{ref,at}}-\overrightarrow{r_{conf,at}}||^2}. \quad (1)$$

Above, ref stands for the reference conformer, conf for the tested conformer, $\overrightarrow{r_{x,at}}$ is the position of an atom in space, $N_{atoms}$ is the number of atoms in the protein. Computing all pairwise RMSDs results in a $N\times N$ matrix, where the components of the matrix are the RMSDs. At this step, we only considered $C_\alpha$ atoms and computed the RMSDs using the Pteros software library[65].

As the second step, a $N\times N$ matrix of $P$ values is calculated for the same pairs of structures. The $P$ values in this matrix correspond to the probability that for a given conformer, the deviation from the experimental FRET distances could be higher than they are (by chance). To compute the $P$ values, first, a weighted sum of squared deviations, $\chi^2$, is calculated. $\chi^2$ is calculated using the set of FRET distances determined by simulations on the reference structure, $\{R_{ref}^{(i)}|1\le i\le N_{measurements}\}$ and a set of distances determined by simulations on the tested structure, $\{R_{conf}^{(i)}|1\le i\le N_{measurements}\}$ with associated estimated uncertainties, $\Delta R_{ref}$, of the reference structure. To compute the set of $P$ values that are later used to calculate the $\langle\langle RMSD\rangle\rangle$, we start by calculating the pairwise values $\chi_{conf,ref}^2$ to later test whether the structure of a conformer can be distinguished from the reference structure. $\chi_{conf,ref}^2$ is computed by:

$$\chi^2 = \chi_{conf,ref}^2 = \sum_{i=1}^{N_{measurements}}\left(\frac{R_{conf}^{(i)}-R_{ref}^{(i)}}{\Delta R_{ref}^{(i)}}\right)^2, \quad (2)$$

where $R_{conf}^{(i)}$ is the FRET distance determined for the FRET pair $i$ on a conformational model conf, $R_{ref}^{(i)}$ is the corresponding distance in the reference conformer, and $\Delta R_{ref}^{(i)}$ is the expected experimental error. The number of FRET measurements, $N_{measurements}$, and the number of independent relevant coordinates (parameters), $N_{fit.param}$, determine the number of degrees of freedom, $N_{dof}$:

$$N_{dof} = N_{measurements} - N_{fit.param.} \quad (3)$$

See also section "Estimation of the complexity of the structure model", for a detailed discussion. For every conformer pair, we calculate a $P$ value or a probability that a sample $\chi^2$ will be larger than $\chi_{conf,ref}^2$:

$$p_{conf,ref} = p(\chi_{conf,ref}^2, N_{dof}) = \int_{\chi_{conf,ref}^2}^{+\infty}f_{N_{dof}}(\chi^2)d\chi^2. \quad (4)$$

$f_{N_{dof}}(\chi^2)$ denotes the chi-squared distribution:

$$f_{N_{dof}}(\chi^2) = \frac{1}{2^{N_{dof}/2}\Gamma(N_{dof}/2)}(\chi^2)^{N_{dof}/2-1}e^{-\chi^2/2}. \quad (5)$$

$\Gamma$ is the Gamma function.

As the third and last step, $\langle\langle RMSD\rangle\rangle$ is computed as a weighted average over the RMSD matrix using the respective $P$ values as weights. $\langle\langle RMSD\rangle\rangle$ is a double average over all reference conformers as well as all conformers being tested:

$$\langle\langle RMSD\rangle\rangle = \frac{1}{N_{conf}}\sum_{ref=1}^{N_{conf}}\frac{\sum_{conf=1}^{N_{conf}}p_{conf,ref}RMSD_{conf,ref}}{\sum_{conf=1}^{N_{conf}}p_{conf,ref}}. \quad (6)$$

**FRET screening.** To assess how well a given structure model or structural ensemble agrees with FRET experiments, we calculate the $\chi^2$ value for each structure in the ensemble. To do that, we need to estimate FRET observables corresponding to the specified conformer. We achieve this by simulating the AV of

the fluorophore attached to a protein by a flexible linker[12] (see Supplementary Note 2).

Often, the reduced chi-squared $\chi_r^2$, also known as chi-squared per degree of freedom, is used to compare different conformers from the same model and as an absolute quality parameter:

$$\chi_r^2 = \chi^2/N_{dof}. \quad (7)$$

However, when models with a different small number of degrees of freedom $N_{dof}$ (e.g., due to different flexibility) are compared, a constant confidence level corresponds to different values of $\chi_r^2$ (Supplementary Fig. 2). Therefore, the use of $\chi_r^2$ for comparing models with different $N_{dof}$ is inconvenient. To overcome this, we introduce an alternative metric, the normalized chi-squared $\chi_{n,68\%}^2$, which equals to 1 for $P = 1 - 68\%$ (one sigma) by definition, independent of the $N_{dof}$ value (Eq. (8), Supplementary Fig. 2). $\chi_n^2$ behaves similarly to $\chi_r^2$, and, in addition, allows to compare several models with a different number of degrees of freedom. Alternatively, a $P$ value can be used directly for this purpose.

$$\chi_n^2 = \chi_{n,68\%}^2 \equiv \chi^2/\text{Inv}.\chi^2(P = 1 - 0.68, N_{dof}), \quad (8)$$

where

$$\text{Inv}.\chi^2(P, N_{dof}) = \frac{2^{-N_{dof}/2}}{\Gamma(N_{dof}/2)}P^{-N_{dof}/2-1}e^{-1/(2P)} \quad (9)$$

is the inverse chi-squared distribution. To visualize the uncertainty of the generated structural models, we display conformations ensembles on two-dimensional plots (Supplementary Fig. 3).

Given an ensemble of structure models, $\chi_n^2$ can be calculated for each conformer. Structures that show better agreement with FRET data have lower $\chi_n^2$. If the FRET-selected ensemble ($\chi_n^2 < 1$) is too diverse (e.g., RMSD > 3 Å), extra FRET pairs can be selected and measured to improve the conformational uncertainty (see below). In this benchmark, reference FRET data ($R_{ref}^{(i)}$, $\Delta R_{ref}^{(i)}$) were determined from the experiment for T4 lysozyme and simulated for other benchmarked proteins using the "true" crystal structure conformations[16]. Structures of T4 lysozyme and its homologs from the PDB were screened against the experimental datasets C1 and C2 in order to select reference conformations for each state (Supplementary Fig. 6). As a result, PDB ID 172L appears to correspond to C1, and PDB ID 3GUN was selected for C2.

**Workflow: selection of a set of optimal FRET pairs.** To maximize the precision of FRET-assisted protein structure determination with a limited number of distance measurements, we developed a method for automated determination of the most informative labeling sites and donor–acceptor (DA) pairs. We define a set of pairs to be the most informative if it leads to the lowest expected model uncertainty, i.e., the lowest possible $\langle\langle RMSD\rangle\rangle$ (Eq. (6)), of a structure model. To find such an optimal set, we compare three different feature selection algorithms (Supplementary Fig. 10): greedy forward selection (Supplementary Note 4), greedy backward elimination (Supplementary Note 5), and an algorithm based on mutual information and inspired by a minimum redundancy maximum relevance (mRMR) algorithm[66] (Supplementary Note 6).

FRET pairs are selected among the full set of all possible pairwise combinations of available labeling sites. Labeling sites can be excluded from calculations based on additional prior information provided by the user, e.g., accessibility, chemical nature and influence on function and stability as determined from mutation analysis or sequence coevolution data. For the proof-of-principle study with the simulated data, we assume that these effects are negligible. However, considering the experimental datasets of T4L, care was taken to avoid these problems. For T4L, automated FRET-pair selection was performed from only 33 FRET pairs as opposed to theoretically possible $162^2/2$ residue-residue combinations, because these 33 pairs were earlier chosen by authors for a functional study of T4L[27] (see the section "T4L: T4 lysozyme site-specific mutation, purification, and labeling"). Despite this low number of available FRET pairs to choose from, only a minor decrease in expected model uncertainty was observed as compared to other proteins (Supplementary Fig. 7), since the 33 pairs were not randomly selected, but handpicked as potentially informative.

In greedy forward feature selection (Supplementary Fig. 1 and Supplementary Note 4), in the first iteration, $\langle\langle RMSD\rangle\rangle$ is calculated for each possible DA pair, and that pair is selected for the DA set that yields the minimal $\langle\langle RMSD\rangle\rangle$. In the next iterations, DA pairs remaining from the previous iteration are probed against the DA set to determine which one leads to the largest decrease in $\langle\langle RMSD\rangle\rangle$; that DA pair is then added to the DA set. The algorithm stops when the desired $\langle\langle RMSD\rangle\rangle$ is reached. Therefore, for conformational ensembles <100,000 structures, the current implementation converges in less than a day on a 4-core desktop computer.

In greedy backward elimination, in the first iteration, $\langle\langle RMSD\rangle\rangle$ is calculated for DA sets containing all possible DA pairs but one. That pair is eliminated for which the remaining DA set yielded the smallest $\langle\langle RMSD\rangle\rangle$; the remaining DA set is narrowed further in an iterative manner (Supplementary Note 5). The algorithm needs to run as many iterations as there are DA pairs available and is therefore slower than the greedy selection algorithm. One run of this algorithm for an

ensemble of <10,000 conformers completes in about one day on a 4-core desktop computer in the current implementation.

In the mutual information-based DA-pair selection algorithm, Shannon conditional entropies are calculated for all pairwise combinations of DA pairs. In the first iteration, the DA pair with the highest Shannon entropy is selected. In the next iterations, the DA pair with the highest minimum Shannon conditional entropy with respect to the previous DA pairs is selected (Supplementary Note 6). That way, the DA pair providing the highest amount of new information not provided by the previously selected DA pairs is selected, similar to mRMR[66]. One run of this algorithm for an ensemble of <100,000 conformers completes in about one day on a 4-core desktop computer in the current implementation.

The greedy pair selection algorithm (Supplementary Note 4) shows the lowest $\langle\langle RMSD\rangle\rangle$ at a low number of measurements ($N_{meas.} \lesssim 5$); however, five pairs is rarely enough to achieve $\langle\langle RMSD\rangle\rangle$ of less than 3 Å. The greedy pair elimination algorithm (Supplementary Note 5) yields the lowest $\langle\langle RMSD\rangle\rangle$ except for a low number of measurements ($N_{meas.} \lesssim 5$); however, this algorithm is also the most computationally demanding. The mutual information-based pair selection algorithm (Supplementary Note 6) shows an intermediate behavior between the greedy pair selection and elimination algorithms, and it is an order of magnitude less computationally demanding than the greedy pair elimination algorithm. We use the greedy selection algorithm if the user requests less than six pairs; otherwise, mutual information-based selection is applied. The user can choose to use a greedy elimination algorithm, which is only advisable if the number of conformers in the ensemble is less than 10,000.

**Theory: estimation of the complexity of the structure model.** Estimation of complexity for a structure model that is used in integrative protein structure determination is essential for quantitative accuracy assessment and automated experiment design. We quantify the complexity of a structure model by the number of relevant independent parameters (coordinates, $N_{fit.param.}$) needed to describe the corresponding conformational ensemble to a given uncertainty ($\langle\langle RMSD\rangle\rangle$). If the structure model is simple, $N_{fit.param.}$ can be calculated analytically, for example, for a rigid-body model, $N_{fit.param.} = (N_{bodies} - 1) \times 6 - N_{bonds}$, where $N_{bonds}$ is the number of hard bonds between the rigid bodies in the model. For non-rigid-body models, obtained from other computational tools, an analytical expression for $N_{fit.param.}$ is usually unavailable[24,25]. Examples of such tools are numerous: molecular dynamics simulations (all-atom or coarse-grained), normal mode-based models, homology models, elastic network models, and others. The effect of non-zero $N_{fit.param.}$ is evident when experimental and model distances are compared. Normally, one expects that ~32% of model distances deviate by more than one sigma from the experimental value, which is not the case for FRET-restrained models (Supplementary Fig. 11).

We thus introduce a heuristic approach for automated $N_{fit.param.}$ determination, which requires as an input only the user-provided conformational ensemble. Initially, to obtain an $N_{fit.param.}$ estimate, we start by assuming $N_{fit.param.,0} = 0$, and determine a set of DA pairs needed to describe the conformations within an ensemble with a desired uncertainty $\langle\langle RMSD\rangle\rangle$ employing our DA-pair selection algorithm. Each DA pair can be seen as a coordinate, and the number of DA pairs corresponds to our definition of $N_{fit.param.}$. Second, we use the number of FRET pairs as predicted by the algorithm at the first stage as the true $N_{fit.param.}$ and re-run the pair selection to obtain an estimate for the number of measurements needed for FRET-assisted structure determination, including cross-validation. Thereby, the number of required measurements is always larger than the model's complexity ($N_{fit.param.}$), reflecting that statistical significance can only be properly assigned to an overdetermined model ($N_{dof} > 0$, see Eq. (3)).

For a FRET-restrained structure model (e.g., generated by FRET-guided NMSim or FRET-restrained MD simulations, see below), the same procedure can be used. Presuming that the explored degrees of freedom in the FRET-restrained model cover all FRET restraints, one can conservatively assume $N_{fit.param.} \geq N_{FRET\ restraints}$. If the structure model itself has more fit parameters than there are measurements, these excessive parameters are also fitted (overfitted), but assigned random values. As a workaround, we use additional measurements to verify the overfitted model by cross-validation. In this study, we use $N_{fit.param.} = N_{FRET\ restraints}$ as a complexity estimate for all FRET-restrained models. Hence, FRET-guided structural sampling must be followed by an additional round of pair selection, so that more FRET pairs are measured for cross-validation.

Overall, these approximations apparently lead to good $N_{fit.param.}$ estimates, and further independent measurements do not change $\chi_n^2$ significantly (Fig. 3c). Reliability of $N_{fit.param.}$ estimates is also evident from the correlation between $\chi_n^2$ and accuracy against the target structure (Supplementary Fig. 7).

**Modeling: unbiased conformation sampling by NMSim.** Structural ensembles unbiased by experimental FRET data were generated by the NMSim software[26]. Ten independent and unbiased NMSim simulations generating 10,000 conformations each were performed, starting from the initial structure and using default parameters for the sampling of large-scale motions. These trajectories are clustered and serve as initial candidates. NMSim is a normal mode-based geometric simulation approach for multiscale modeling of protein conformational changes that incorporate information about preferred directions of protein motions into a geometric simulation algorithm. NMSim follows a three-step protocol: in the first step, the protein structure is coarse-grained by the software FIRST[67] into rigid parts connected by flexible links[68]. In the second step, low-frequency normal modes are computed by rigid cluster normal mode analysis (RCNMA)[33]. In the third step, a linear combination of the first ten normal modes was used to bias backbone motions along the low-frequency normal modes, while the side-chain motions were biased toward favored rotamer states. A detailed list of used simulation parameters is given in the Supplementary Note 7.

**Modeling: FRET-guided NMSim.** To improve the sampling of the conformational space in regions most relevant according to the experiment, we extended the NMSim approach by a Markov Chain Monte Carlo step to prioritize conformations lying in such regions (Supplementary Fig. 4). In every NMSim iteration, the generated conformation is scored with respect to its agreement with experimental data using the $\chi_n^2$ metric. Then, according to the Metropolis–Hastings approach,

$$P_{accept} = \exp\left(\frac{\chi_{n,previous}^2 - \chi_{n,current}^2}{kT}\right) \quad (10)$$

is computed, and the conformation is accepted and used in the next NMSim iteration if $P$ is larger than a uniformly distributed random number sampled from the range [0, 1]; else, the conformation is discarded, and the previous one is used in the next NMSim iteration. As a result, conformations are generated that are both stereochemically plausible and agree with experimental data. To improve the sampling and enable the exploration of multiple local minima, an annealing procedure is applied in which $kT$ is varied from almost 0 to 1 units of $\chi_n^2$ and back to 0 (see Supplementary Note 7). A single FRET-guided NMSim simulation contains two such annealing cycles. If models with good FRET agreement ($\chi_n^2 \to 1$) cannot be obtained from FRET-guided simulations, alternative seed structures should be considered.

Several models with different levels of detail are available in the FRET suite to describe the dye in coarse-grained simulations. Besides the standard AV model[69], we implemented two more detailed dye models: (i) accessible and contact volume (ACV) model[16,70], and (ii) the FP model where a fluorescent protein is fused to the studied protein by a flexible peptide linker[21]. In the ACV model, the occupancy of the layer near the protein surface is calibrated from time-resolved fluorescence anisotropy measurements, reflecting the sticking of the fluorophore and the corresponding increase in residual anisotropy. The FP model accounts for the distribution of the end-to-end distance of a flexible polypeptide chain, calibrated by the experimental data and the steric exclusion effects of a big dye, like green fluorescent protein or similar.

**Modeling: FRET-restrained MD simulations.** To reconstruct structures to maximum detail, we developed a procedure to incorporate FRET restraints in atomistic molecular dynamics (MD) simulations applying an implicit dye representation (Supplementary Fig. 5). Integration of MD simulations into a FRET-assisted hybrid modeling workflow is especially useful because it allows to recover fine-grained structural detail, which would be otherwise missing from models based on sparse experimental data. These details include, but are not limited to, secondary structure information, side-chain orientations, and fulfillment of steric restrictions.

It was previously demonstrated, how information from FRET measurements can be incorporated directly in MD simulations, by applying FRET restraints between explicit flexibly tethered dyes[13,71]. In our approach, FRET restraints are applied to the mean dye position, as opposed to the explicit dye. Unlike the immediate position of a fluorophore, its mean position with respect to the local backbone does not change as easily. When FRET restraints are applied to explicitly modeled fluorophores directly, the flexible dye linker becomes a soft entropic spring[72] and absorbs most of the strain. In our method, we partially mitigate complications of explicit dye simulations, such as potential inaccuracies of dye force field parametrizations and large convergence times (>100 ns[41]) of fluorophore diffusion. Finally, FRET observables determined in experiment have a statistical nature: they represent state-specific ensemble averages and underlying distributions, rather than immediate quantities. Therefore, application of "statistical" FRET restraints to pseudoatoms that are constructed to mimic statistically averaged fluorophore positions is more straightforward and effective. The statistical representation of dyes by pseudoatoms in a rigid-body docking approach was successfully used by Brunger and coworkers and us[12,73]. However, to our knowledge, it was not demonstrated for all-atom MD simulations.

To generate the restraints, first AV calculations are performed for each labeling position. Second, pseudoatoms are positioned at the mean position of every AV. These pseudoatoms do not interact with protein or solvent atoms. To keep the pseudoatoms in their initial positions relative to the corresponding part of the backbone, harmonic restraints are used: pseudobonds are created between the pseudoatom and $C_\alpha$ and $C_\beta$ atoms of amino acids up to two residues toward the C- or N-termini of the protein from the amino acid, where the fluorophore linker is attached. Thus, each pseudoatom is anchored to ten nearby backbone atoms. The positions of pseudoatoms, the anchoring bonds, and FRET restraints are recalculated every 2 ns during the simulation to account for changes in local structure.

To mimic the measured FRET distances, pseudoatoms are restrained with respect to each other using harmonic-linear restraints. If the distance between pseudoatoms corresponds exactly to the measured donor–acceptor distance, no additional force is applied to pseudoatoms. To prevent unphysical unfolding of the protein, the FRET-restraint force is capped at an empirically determined value $F_{max} = 50$ pN, which is reached when the bond length ($R_{DA}$) is more than one standard error ($R_{exp}$) away from the optimum ($R_{exp}$, Supplementary Fig. 5d). The error for each FRET distance is determined from experimental data. Force constants for each FRET-restraint are tuned such that for every pseudoatom the magnitude of the total FRET-restraints vector is $\leq F_{max}$, resulting in force constants for FRET restraints in the range from 0.7 to 14 pN/Å, depending on their collinearity. Force constants of the pseudobonds that attach pseudoatoms to their local backbone atoms are set ten times higher than those for FRET restraints. FRET restraints are implemented using the AMBER interface for NMR restraints ("DISANG" file).

The AMER16 suite of molecular simulation codes[74] was used to perform MD simulations. All co-crystallized waters and ligands were removed from the crystal structures. Hydrogen atoms were removed and re-added by tleap[75] from the AMBER Tools suite. The TIP3P explicit water model[76] was used to solvate proteins in a periodic truncated octahedral box with at least 12 Å of solvent in every direction from the protein surface. Sodium and chloride counter ions were added to neutralize the systems. MD simulations were performed with the ff14SB force field[77] using the GPU version of pmemd[78]. The SHAKE algorithm[79] was used to constrain bond lengths of hydrogen atoms. Long-range electrostatic interactions were evaluated using the particle mesh Ewald method[80]. Hydrogen mass repartitioning[81] and a time step of 4 fs were used. A five-stage equilibration procedure was pursued: First, 100 steps of steepest descent and 400 steps of conjugate gradient minimization were performed, while solute atoms were restrained to their initial positions by harmonic restraints with force constants of 5 kcal mol$^{-1}$ Å$^{-2}$. Second, the temperature of the system was raised from 100 K to 300 K in 50 ps of NVT-MD simulations. Third, 150 ps of NPT-MD simulations were performed to adjust the system density. Finally, the force constants of harmonic restraints were gradually reduced to zero during 2 ns of NVT-MD simulations. Production NVT-MD simulations were carried out at 300 K, using the Berendsen thermostat[82] and a coupling constant of 0.5 ps. Three independent replicas of MD simulations (1 μs per simulation) were performed for each system using different random number generator seeds to assign initial velocities.

**T4L: T4 Lysozyme site-specific mutation, purification and labeling**. T4L site-directed mutagenesis was performed on the cysteine-less pseudo-wild-type encoded backbone using the pET11a (Life Technologies, Corp) vector using standard methods[83–85]. A list of primer used for mutagenesis is provided in Supplementary Table 2. For protein expression and purification, the plasmid containing T4L desired mutations (an unnatural amino acid –p-acetyl-L-phenylalanine or pAcPhe, in the N-terminal subdomain (NTsD) and the replacement to a Cys in the C-terminal subdomain (CTsD)) was co-transformed with pEVOL[84] into BL21(DE3) E. coli strains (Life Technologies Corp.). Transformed E. coli were plated onto LB-agar plates supplemented with ampicillin and chloramphenicol for single-colony selection. For each variant, a single colony was inoculated into 100 mL of LB with antibiotics and grown overnight at 37 °C in a shaking incubator, followed by inoculation of a 1 L of LB medium supplemented with the respective antibiotics and 0.4 g/L of pAcPhe (SynChem) with 50 mL of the overnight culture. The culture was grown at 37 °C until an OD600 of 0.5 was achieved. The protein production was induced for 6 h by addition of 1 mM IPTG and 4 g/L of arabinose. Harvested cells were lysed in 50 mM HEPES, 1 mM EDTA, and 5 mM DTT pH 7.5 and purified using a monoS 5/5 column (GE Healthcare) with an eluting gradient from 0 to 1 M NaCl according to standard procedures. High-molecular-weight impurities were removed by passing the eluted protein through a 30-kDa Amicon concentrator (Millipore), followed by subsequent concentration and buffer exchange to 50 mM PB, 150 mM NaCl pH 7.5 of the protein flow through with a 10-kDa Amicon concentrator.

Site-specific labeling of T4L was accomplished using orthogonal chemistry following manufacturer suggestion. For labeling, the Keto functional group of pAcPhe at the NTsD, the Alexa 488 with hydroxylamine linker chemistry was used (Life Technologies Corp.). Cysteine sites were labeled via a thiol reaction with maleimide linkers of Alexa-647. FRET or DA variants were labeled sequentially—first thiol and second the keto handle[85]. A proper donor-only reference sample was only kept before proceeding with the acceptor labeling. The selected FRET pair has a Förster distance $R_0$ of 52 Å.

**T4L: FRET experiments and analysis**. To resolve the conformational heterogeneity of T4L, donor-only and FRET-labeled T4L variants were studied by time-resolved fluorescence spectroscopy using Time Correlated Single Photon Counting (TCSPC) and single-molecules studies with confocal multi-parameter fluorescence detection.

Donor-only and FRET-labeled T4L variants were measured by TCSPC using either an IBH-5000U (IBH, Scotland) or a Fluotime200 (Picoquant, Germany) system. The excitation source of the IBH machine was a 470-nm diode laser (LDH-P-C470, Picoquant, Germany) operating at 10 MHz for donor excitation and a 635 nm (LDH-P-C635, Picoquant, Germany) for acceptor excitation. The excitation

and emission slits were set to 2 nm and 16 nm, respectively. The excitation source of the Fluotime200 system was a white light laser (SuperK extreme, NKT Photonics, Denmark) operating at 20 MHz for both donor (485 nm) and acceptor (635 nm) excitation with excitation and emission slits set to 2 nm and 5 nm, respectively. Additionally, in both systems, cut-off filters were used to reduce the amount of scattered light (>500 nm for donor and >640 nm for acceptor emission).

For green detection, the monochromator was set to 520 nm and for red detection to 665 nm. All measurements were conducted under magic-angle conditions (excitation polarizer 0°, emission polarizer 54.7°, VM), except for anisotropy where the position of the emission polarizer was alternately set to 0° (VV) or 90° (VH).

In the IBH system, the TAC-histograms were recorded with a bin width of 14.1 ps within a time window of 57.8 ns, while the Fluotime200 was set to a bin width of 8 ps within a time window of 51.3 ns. The average number of collected photons per sample was $30 \times 10^6$ photons.

A global joint analysis of the donor-only and FRET-labeled samples was implemented in order to assure proper donor reference samples, determination of the mean interdye distances, $\langle R_{DA} \rangle$, and assignment of states by sharing the population parameters on the FRET-labeled samples. The analysis and justification of the methods are reported in Sanabria et al.[27]. In short, the donor-only labeled samples were fit with a multiexponential decay model (Eq. (25), Peulen et al.)[41]. All FRET-induced donor VM decays were fit using the corresponding donor-only decay parameters with a sum of Gaussian distributed states to derive $\langle R_{DA} \rangle$. By using global analysis, we assure conformational states are assigned via the linked population fractions. A 2σ statistical uncertainty and an error propagation rule considering the uncertainty in fluorophore orientation ($\kappa^2$ error) was used to consider the overall uncertainty ($+/-$ err). The derived distances for two states are presented in Supplementary Table 1. The error estimation considers: (i) upper estimates for the uncertainty of the orientation factor[69], $\kappa^2$, (ii) statistical uncertainties of the analysis[41], (iii) estimates for systematic errors due to imprecise reference samples[41], and (iv) uncertainty estimates for modeling the spatial distribution of the dyes based on the dye's residual anisotropies[13] (see Supplementary Table 3).

**Reporting summary**. Further information on research design is available in the Nature Research Reporting Summary linked to this article.

## Data availability
Data supporting the findings of this paper are available from the corresponding authors upon reasonable request. A reporting summary for this article is available as a Supplementary Information file. The original experimental data supporting the findings in this work are available from Zenodo (https://doi.org/10.5281/zenodo.3376527). Structure models of the T4L based on experimental FRET restraints were deposited to PDB-dev (PDB-dev ID: PDBDEV_00000044) using the FLR-dictionary extension (developed by PDB and the Seidel group) available on the IHM working group GitHub site (https://github.com/ihmwg/FLR-dictionary). Source data are provided with this paper.

## Code availability
Our software suite FPS 2.0 for automated FRET-assisted modeling relies upon four software tools: (1) "Olga", a program for FRET screening and optimal FRET-pair selection (experiment planning): https://github.com/Fluorescence-Tools/Olga. (2) "NMSim", a coarse-grained geometric simulations software for unrestrained conformational sampling and FRET-guided coarse-grained modeling: http://nmsim.de. (3) "FRETrest" command-line tool for FRET-restrained molecular dynamics simulations: https://github.com/Fluorescence-Tools/FRETrest. (4) "LabelLib", a command-line tool for simulations of dye accessible volumes and interdye distance distributions: https://github.com/Fluorescence-Tools/labellib.

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

## Acknowledgements

This work was funded in part by the German Research Foundation (DFG) within the Collaborative Research Center SFB 1208 "Identity and Dynamics of Membrane Systems—From Molecules to Cellular Functions" (T.P. A03 to H.G. and T.P. A08 to C.S.) and by the European Research Council through the Advanced Grant 2014 (number 671208) to C.S. H.S. acknowledges support of the Alexander von Humboldt, NSF-BIO 1749778, NIH 1P20GM121342-01A1. We are grateful for computational support and infrastructure provided by the "Zentrum für Informations- und Medientechnologie" (ZIM) at the Heinrich Heine University Düsseldorf and the computing time provided by the John von Neumann Institute for Computing (NIC) to H.G. on the supercomputer JURECA at Jülich Supercomputing Centre (JSC) (user IDs: HKF7, HDD20).

## Author contributions

M.D. developed methods and performed programming and computations. H.S., D.R., K.H., and T.P. designed T4L FRET network and prepared, labeled, and analyzed experimental data for T4 Lysozyme. M.D., C.S., and H.G. analyzed the data, discussed the results, and wrote the paper. C.A.H. prepared data files for submission to PDB-Dev. C.S. and H.G. performed the study design and supervised the project.

## Funding

## Competing interests

The authors declare no competing interests.
