## [Peer Review File · Nature Communications]

Reviewers' Comments:

Reviewer #1:

Remarks to the Author:

This manuscript unifies a set of tools for using FRET measurements to guide modeling of biomolecular structures that these authors have been developing for some time. These integrative structural biology approaches are gaining momentum and this is a timely topic. FRET has particular strengths that make it an appropriate experimental tool for many systems, especially flexible and dynamic systems.

This study focuses on a set of test proteins, each with 2 distinct configurations. Most of the test proteins have both structures for both configurations in the pdb with the exception of one from the CASP11 project (which has more available) and another protein which is from their own experiments. The authors use one of the configurations as a starting point, simulate FRET data from the protein (with exception of the one experiment on T4 lysozyme), generate ensembles of computed configurations, and then use the FRET data to predict the other real structure from the ensemble. They also present tools for optimizing the number of FRET pairs required and their location for good convergence of the method.

Their results are good for their test set of proteins with FRET restraints guiding selection of real structures from the ensemble of computationally generated structures.

Some strengths of the paper include the completeness of the set of tools, the emphasis on error analysis/propagation, and the validation of the final models. In addition, although many parts of the work flow are previously published by the group (especially their reference 16), this paper presents a complete solution to the problem. I can support publication. I have a few specific comments for the authors.

Specific Comments:

1. The authors provide no discussion of how handle the presumed two FRET values derived from each of the two interconverting structures for each protein. Further concerning, would be systems with more complex dynamics than switching between two configurations. These cases could produce very complex FRET signals at single molecule level, but would average to some value in ensemble FRET. In the introduction, the authors suggest applicability to many different sorts of dynamics, so this issue would suggest more than 2 states might be expected.
2. The title, abstract and introduction suggest no distinction between ensemble and single molecule FRET for use with this computational method. With multiple conformers, like in the previous comment, ensemble FRET signals will average the two and the authors should expand how they imagine using such measurements in the workflow.
3. On Line 228 – the authors mention they simulated the FRET data by a published method (they cite reference 16, their 2016 Current Opinion paper). I was curious about this method, so looked in the 2016 paper. The data simulation method in that paper cites back to a 2012 ChemPhysChem paper. I gave up trying to understand the data simulation method at that point and still don't have a clear idea of how they simulate fret data. Maybe a brief description of that method (supplemental note) for the current paper would be appreciated by readers.
4. On Line 317 – It seems strange to deposit models in PDB-dev that are built by simulating data from models pulled from the PDB. I don't quite get the significance of these new models.
5. On Line 325 – The authors mention the potential of suing this method to model intrinsically disordered proteins. They might also alert the readers to the fact that extracting real distance information from FRET measurements of disordered proteins require non-trivial modeling of the random motion that results in averaging of rapidly fluctuating configurations.

Minor points:

- a. On Line 63 and line 574 – 600, and in suppl. Fig 4: Regarding the implicit dye model and restraints on pseudoatoms: Brunger, among others, has previously guided docking simulations by single molecule FRET signals modeled as restraints from rigidly attached pseudoatoms at the average location of the dye from MD simulation of the dye position. (A.T. Brunger et al., Journal of Structural Biology 173 (2011) 497–505)
- b. On Line 310: there is a typo “we b FRET-derived structural models.”
- c. In Supplementary figure 5, panels b and c were missing in my document.

Reviewer #2:

Remarks to the Author:

Dimura and colleagues present a set of new software tools for designing FRET experiments, using the experimental results for structural modeling, and assessing the quality of the model, including careful error analysis. Even though this type of approach has now been employed in the community for quite some time, its use has largely been limited to specialists that develop the required software themselves. Making such modeling software more widely available is thus a key step and will be of great interest to many practitioners in the field (and beyond, since the tool can almost directly be used for related techniques such as EPR). The manuscript is very well written, and careful documentation for the software tools is provided. I thus strongly support publication in Nature Communications essentially as is.

Just a few minor points that the authors may want to consider:

Line 121: “...we answer the questions how many and which pairs...” should be “...we answer the questions of how many and which pairs...”

Line 310: last subclause seems to be scrambled.

Line 322: “The programs for structural modeling in the suite can be applied to [...] biomolecules with disordered regions, as long as structural knowledge of disordered regions is available in the prior.”

It would be interesting to specify what “structural knowledge of disordered regions” means in this context, since many disordered proteins or regions have no well-defined secondary or tertiary structure. The authors may want to cite the following summary of the current state of modeling disordered proteins based on experimental data (especially FRET) with different approaches: <https://www.ncbi.nlm.nih.gov/pubmed/30471690>

Reviewer #3:

Remarks to the Author:

Dimura et al. present an interesting study in which the authors explore the precision and accuracy with which ensemble and/or single-molecule FRET experiments can provide detailed structural information on proteins. Trigger for this study has been the increasing use of integrative approaches to gain structural insights into protein structure and function, particularly for intrinsically disordered proteins. A significant drawback of FRET-based methods is the small number of FRET-pairs that can typically be used in such approaches. While this is less of a problem for determining overall dimensions and ‘shapes’ of disordered protein ensembles, it becomes a limiting factor if the goal is to determine the structure of a folded protein. Here, the authors introduced a multi-step procedure (i) to

determine the most informative FRET pairs for structure determination and (ii) to develop quality measures to assess the precision of the obtained structures. Since the work of Dimura et al. is a benchmark study, the authors chose model proteins with known target structures such that they were also able (iii) to assess the accuracy of the obtained structures. Since even the large number of 20 FRET pairs is insufficient to determine the structure of a protein unambiguously, the routine presented by Dimura et al. requires prior knowledge. This knowledge either comes from structures of alternative states of a protein (x-ray, NMR, cryo-EM) or, if structures are unavailable, from educated guesses, i.e., modeling by various approaches. These initial guesses are randomized using different methods (NMSim, MD) to cover a larger conformational space. Key of the whole approach is to determine labeling positions for the FRET-dyes that allow an optimal discrimination between conformers in the initial ensemble. The authors suggest an iterative procedure to identify FRET-pairs that minimize the quantity $\langle \langle \text{RMSD} \rangle \rangle$, which is a weighted RMSD across all conformer pairs in the ensemble. Using six proteins with simulated FRET-observables (apart from T4 lysozyme for which data had previously been published), the authors finally show that they successfully identify the correct fold with remarkable RMSD values of $\sim 3\text{\AA}$. Implementations of FRET-restraint simulations help to fine-tune the best structure in the initial ensemble. This is achieved by fixing the mean positions of the dyes while they are connected to the protein backbone via springs (spring-constant depends on the error of the FRET-measurement and by a total force of $\pm 50\text{ pN}$, which, according to the authors, should be low enough to prevent unfolding).

In summary, the paper describes a technically sound procedure to generate best-fit structures based on FRET experiments, including a search algorithm for determining the number and type of most discriminative FRET pairs. Although demanding, I think the procedure is a good step towards using FRET-experiments for protein structure determination. However, as explained by the authors in their introduction, the tool would be most interesting for multi-domain proteins, complexes, dynamic systems with unstructured regions, and systems with lowly populated states. The choice of the proteins, all well folded, is ideal to demonstrate the accuracy of the method (in the end, it's a method description), but it fails to demonstrate its potential for the more complicated cases such as partially disordered proteins. In addition, for disordered systems in particular, less demanding integrative approaches, e.g., Bayesian reweighting, are in use since several years. The paper is very sound and I only have a few technical questions

1. It seems to me that determining the correct secondary structure is an obvious issue. The authors also note that if the initial ensemble does not contain structures with the correct secondary structure, it will be difficult to find the right structure (p 7 line 202). I am not convinced that this scenario can be easily picked up by a large χ^2 value, unless the number of measurements increases massively. In other words, what is the number of false-positives?
2. Although doable, the average number of FRET-pairs easily exceeds 20. Do the authors consider this a realistic effort given alternative methods with significantly higher resolution such as cryo-EM? Even a low-resolution structure in cryo-EM (6 – 10 Å) contains more information than 20 FRET pairs.
3. Although the simulated FRET-observables used for five of the proteins (experimental FRET data are published for T4-Lysozyme) are well suited to test the method, the choice of labeling positions often includes constraints (stability, function) beyond that of being most discriminative for structures. The authors mentioned that FPS 2.0 can account for such constraints. Therefore, it would have been great to discuss how many of the FRET pairs for the five proteins are in accord or in conflict with such additional constraints, e.g., amino acid conservation or mutations.
4. Fig. 7 seems to miss information that is described in the caption. I didn't find the scatter plots in the upper left corner.
5. Finally a general remark: While structure is important to understand protein function, this understanding often requires knowledge of the precise arrangement of side chains, clearly information that cannot be provided by a low-resolution technique such as FRET. For less detailed problems, e.g., overall domain arrangements or assemblies of complexes, I'd guess that also a less detailed FRET-analysis would be adequate.

NCOMMS-20-04355: Answers to the reviewers

Automated and optimally FRET-assisted structural modeling

Mykola Dimura^{1,2}, Thomas-O. Peulen¹, Hugo Sanabria^{1,3}, Dmitro Rodnin¹, Katherina Hemmen¹, Christian A. Hanke¹, Claus A.M. Seidel^{1,*}, Holger Gohlke^{2,4,*}

We thank the reviewers for taking the time to read our manuscript carefully. We took their questions and concerns very seriously and fully addressed them in the revised version of our manuscript. Your questions definitely helped us to improve the quality of our manuscript. All major scientific changes in the text are highlighted in yellow. In addition we tried to improve the readability of the text.

Rev#1

This manuscript unifies a set of tools for using FRET measurements to guide modeling of biomolecular structures that these authors have been developing for some time. These integrative structural biology approaches are gaining momentum and this is a timely topic. FRET has particular strengths that make it an appropriate experimental tool for many systems, especially flexible and dynamic systems.

This study focuses on a set of test proteins, each with 2 distinct configurations. Most of the test proteins have both structures for both configurations in the pdb with the exception of one from the CASP11 project (which has more available) and another protein which is from their own experiments. The authors use one of the configurations as a starting point, simulate FRET data from the protein (with exception of the one experiment on T4 lysozyme), generate ensembles of computed configurations, and then use the FRET data to predict the other real structure from the ensemble. They also present tools for optimizing the number of FRET pairs required and their location for good convergence of the method.

Their results are good for their test set of proteins with FRET restraints guiding selection of real structures from the ensemble of computationally generated structures.

Some strengths of the paper include the completeness of the set of tools, the emphasis on error analysis/propagation, and the validation of the final models. In addition, although many parts of the work flow are previously published by the group (especially their reference 16), this paper presents a complete solution to the problem. I can support publication. I have a few specific comments for the authors.

Specific Comments:

1. The authors provide no discussion of how handle the presumed two FRET values derived from each of the two interconverting structures for each protein. Further concerning would be systems with more complex dynamics than switching between two configurations. These cases could produce very complex FRET signals at single molecule level, but would average to some value in

ensemble FRET. In the introduction, the authors suggest applicability to many different sorts of dynamics, so this issue would suggest more than 2 states might be expected.

Detailed description of the strategies for separating state specific information from experimental FRET data is a point of great interest, but falls outside of the scope of the current manuscript. Application of leveraging time-resolved experiments to resolve states was described in more detail in ref. {Peulen, Opanasyuk 2017}. Briefly, state-specific inter-dye distance distributions can be reliably identified by fluorescence decay analysis of sub-populations for states with exchange relaxation times of microseconds to infinity. Attribution of distance distributions to specific states can be done based on species amplitudes. For states that exchange at timescales of 100 microseconds and longer, several methods exist for quantitative separation of states from single-molecule photon traces {dynPDA, hMC, etc}.

We added this information to the Discussion section on p. 14.

2. The title, abstract and introduction suggest no distinction between ensemble and single molecule FRET for use with this computational method. With multiple conformers, like in the previous comment, ensemble FRET signals will average the two and the authors should expand how they imagine using such measurements in the workflow.

We suggest to prevent the information loss from averaging of the FRET signals by using fluorescence decay analysis of time-resolved (sub)ensemble FRET data, as detailed in more detail in {Peulen, Opanasyuk 2017}. We elaborate on this topic in the Discussion section on p. 14.

3. On Line 228 – the authors mention they simulated the FRET data by a published method (they cite reference 16, their 2016 Current Opinion paper). I was curious about this method, so looked in the 2016 paper. The data simulation method in that paper cites back to a 2012 ChemPhysChem paper. I gave up trying to understand the data simulation method at that point and still don't have a clear idea of how they simulate fret data. Maybe a brief description of that method (supplemental note) for the current paper would be appreciated by readers.

We calculate average inter-dye distances from AV simulations and the crystal structure of the target state. We compute the error of average inter-dye distance by propagating the error of Efficiency $\Delta E=0.06$. This is the typical value of an error for single-molecule FRET measurements as determined in a community-wide benchmark study {Hellenkamp et al. 2018}. The effect of dye diffusion quenching by the dye's local environment for time-resolved fluorescence is discussed in detail in ref. {Peulen, Opanasyuk 2017}. A corresponding statement has been added in the section "Benchmarking the methodology" (p. 8).

4. On Line 317 – It seems strange to deposit models in PDB-dev that are built by simulating data from models pulled from the PDB. I don't quite get the significance of these new models.

It is the policy of PDB-dev to accept only new structure models.

We show that it is possible to create an accurate conformational model of a protein based on a crystal structure of a different structural state. We only deposit the models generated by FRET-guided modeling with experimentally-obtained data for T4 lysozyme. We used experimental FRET data for state C2 to generate the structure starting from PDB-ID 172L and arrive at a conformation similar to PDB-ID 148L, without any information from 148L. We also do the opposite and use experimental FRET-data for C1 and the seed structure with PDB-ID 148L; as a result, we arrive close to PDB-ID 172L, even though the modelling had no information about the PDB-ID 172L.

5. On Line 325 – The authors mention the potential of using this method to model intrinsically disordered proteins. They might also alert the readers to the fact that extracting real distance information from FRET measurements of disordered proteins require non-trivial modeling of the random motion that results in averaging of rapidly fluctuating configurations.

Indeed, rapid fluctuations of disordered systems cause signal averaging, mandating non-trivial modelling. We added a corresponding note in the Discussion section (p. 16).

Minor points:

a. On Line 63 and line 574 – 600, and in suppl. Fig 4: Regarding the implicit dye model and restraints on pseudoatoms: Brunger, among others, has previously guided docking simulations by single molecule FRET signals modeled as restraints from rigidly attached pseudoatoms at the average location of the dye from MD simulation of the dye position. (A.T. Brunger et al., *Journal of Structural Biology* 173 (2011) 497–505)

Indeed, Brunger et al. used rigidly attached pseudoatoms in a rigid body-based modelling. We used a very similar approach in our previous study {Kalinin 2012}. However, it is uncommon to use a statistical representation of the dye in all-atom molecular dynamics (MD) simulations, since the dye can be naturally added explicitly. In our opinion, the combination of a coarse-grained dye representation and all-atom simulations provides better control for FRET-guided MD. We refer to the innovative work by Brunger et al. in the subsection “FRET-restrained MD simulations” of the Methods section (p. 23).

b. On Line 310: there is a typo “we b FRET-derived structural models.”

Thank you, indeed, the word was accidentally scrambled; we fixed it in the manuscript.

c. In Supplementary figure 5, panels b and c were missing in my document.

Panel labels were missing, we fixed them now.

Rev#2

Dimura and colleagues present a set of new software tools for designing FRET experiments, using the experimental results for structural modeling, and assessing the quality of the model, including careful error analysis. Even though this type of approach has now been employed in the community for quite some time, its use has largely been limited to specialists that develop the required software themselves. Making such modeling software more widely available is thus a key step and will be of great interest to many practitioners in the field (and beyond, since the tool can almost directly be used for related techniques such as EPR). The manuscript is very well written, and careful documentation for the software tools is provided. I thus strongly support publication in *Nature Communications* essentially as is.

Just a few minor points that the authors may want to consider:

Line 121: “...we answer the questions how many and which pairs...” should be “...we answer the questions of how many and which pairs...”

Line 310: last subclause seems to be scrambled.

Yes, we corrected the sentences as suggested.

Line 322: “The programs for structural modeling in the suite can be applied to [...] biomolecules with disordered regions, as long as structural knowledge of disordered regions is available in the prior.”

It would be interesting to specify what “structural knowledge of disordered regions” means in this context, since many disordered proteins or regions have no well-defined secondary or tertiary structure. The authors may want to cite the following summary of the current state of modeling disordered proteins based on experimental data (especially FRET) with different approaches: <https://www.ncbi.nlm.nih.gov/pubmed/30471690>

We added a statement about the special issues of disordered proteins in the Discussion section and refer to the article by Holmstrom et al. (p. 16).

Rev#3

Dimura et al. present an interesting study in which the authors explore the precision and accuracy with which ensemble and/or single-molecule FRET experiments can provide detailed structural information on proteins. Trigger for this study has been the increasing use of integrative approaches to gain structural insights into protein structure and function, particularly for intrinsically disordered proteins. A significant drawback of FRET-based methods is the small number of FRET-pairs that can typically be used in such approaches. While this is less of a problem for determining overall dimensions and ‘shapes’ of disordered protein ensembles, it becomes a limiting factor if the goal is to determine the structure of a folded protein. Here, the authors introduced a multi-step procedure (i) to determine the most informative FRET pairs for structure determination and (ii) to develop quality measures to assess the precision of the obtained structures. Since the work of Dimura et al. is a benchmark study, the authors chose model proteins with known target structures such that they were also able (iii) to assess the accuracy of the obtained structures. Since even the large number of 20 FRET pairs is insufficient to determine the structure of a protein unambiguously, the routine presented by Dimura et al. requires prior knowledge. This knowledge either comes from structures of alternative states of a protein (x-ray, NMR, cryo-EM) or, if structures are unavailable, from educated guesses, i.e., modeling by various approaches. These initial guesses are randomized using different methods (NMSim, MD) to cover a larger conformational space. Key of the whole approach is to determine labeling positions for the FRET-dyes that allow an optimal discrimination between conformers in the initial ensemble. The authors suggest an iterative procedure to identify FRET-pairs that minimize the quantity $\langle\langle\text{RMSD}\rangle\rangle$, which is a weighted RMSD across all conformer pairs in the ensemble. Using six proteins with simulated FRET-observables (apart from T4 lysozyme for which data had previously been published), the authors finally show that they successfully identify the correct fold with remarkable RMSD values of $\sim 3\text{\AA}$. Implementations of FRET-restraint simulations help to fine-tune the best structure in the initial ensemble. This is achieved by fixing the mean positions of the dyes while they are connected to the protein backbone via springs (spring-constant depends on the error of the FRET-measurement and by a total force of ± 50 pN, which, according to the authors, should be low enough to prevent unfolding).

In summary, the paper describes a technically sound procedure to generate best-fit structures based on FRET experiments, including a search algorithm for determining the number and type of most discriminative FRET pairs. Although demanding, I think the procedure is a good step towards using FRET-experiments for protein structure determination. However, as explained by the authors in their introduction, the tool would be most interesting for multi-domain proteins, complexes, dynamic systems with unstructured regions, and systems with lowly populated states. The choice of the proteins, all well folded, is ideal to demonstrate the accuracy of the method (in the end, it’s a method description), but it fails to demonstrate its potential for the more complicated cases such as

partially disordered proteins. In addition, for disordered systems in particular, less demanding integrative approaches, e.g., Bayesian reweighting, are in use since several years. The paper is very sound and I only have a few technical questions

1. It seems to me that determining the correct secondary structure is an obvious issue. The authors also note that if the initial ensemble does not contain structures with the correct secondary structure, it will be difficult to find the right structure (p 7 line 202). I am not convinced that this scenario can be easily picked up by a large χ^2 value, unless the number of measurements increases massively. In other words, what is the number of false-positives?

It is possible to introduce a change in the secondary structure of a protein but preserve the overall arrangement of subdomains, which would be difficult to detect by FRET. However, misassignment of secondary structure introduced by a computational structure prediction tool often propagates to changes in tertiary structure and mutual orientation of subdomains. FRET can detect such changes in subdomain orientation and thus, indirectly, falsify incorrect secondary structure predictions. We now better clarify this issue and add Figure 5, which shows the correlation between the local Distance Difference Test (IDDT) score and FRET χ_n^2 . IDDT score specifically highlights the (dis)similarity of secondary and local structure. We aim to determine the actual rate of potential false-positives by the analysis of CASP targets in the future.

2. Although doable, the average number of FRET-pairs easily exceeds 20. Do the authors consider this a realistic effort given alternative methods with significantly higher resolution such as cryo-EM? Even a low-resolution structure in cryo-EM (6 – 10 Å) contains more information than 20 FRET pairs.

FRET studies with more than 10 and up to hundred pairs are now not entirely uncommon {Hellenkamp HSP90 2017, Kalinin FPS 2012, Sanabria T4L 2020}. For challenging heterogenous structures FRET experiments can provide valuable information.

Resolution in cryo-EM usually means the resolution of experimental data itself, i.e., resolution of the electron density map. The FRET analog would be the uncertainty of inter-dye distance measurements. However, FRET data is sparse, unlike EM maps. The connection between the uncertainty of measured distances and their number on the one hand and the accuracy of FRET-based hybrid models on the other hand is not trivial. We hope, our work helps clarifying this relation. The uncertainty of an EM-based structural model is typically close to the resolution of the EM map, however, the uncertainty of a protein model fitted to an EM map can sometimes exceed the resolution of the map itself, analogous to the FRET-based models.

Here, we demonstrate for the example of six proteins that a small set of FRET measurements can resolve heterogenous systems. In the future and in combination with cryo-EM, FRET experiments can provide kinetic information and support cryo-EM analysis. Moreover, FRET can be performed on smaller molecules and *in vivo*, and thus can be more preferable for some applications. In our approach, low uncertainty on a per-residue level is achieved by the computational modelling tools and restraints, complementing FRET data. Typically, computational modelling tools provide different levels of confidence for different structural parts, which propagates to element-specific variations of the uncertainty of an integrative model. We added Figure 5 to quantify and illustrate this using the IDDT score. Variations of uncertainty for different parts of the structure are also typical for cryo-EM models.

3. Although the simulated FRET-observables used for five of the proteins (experimental FRET data are published for T4-Lysozyme) are well suited to test the method, the choice of labeling positions often includes constraints (stability, function) beyond that of being most discriminative for

structures. The authors mentioned that FPS 2.0 can account for such constraints. Therefore, it would have been great to discuss how many of the FRET pairs for the five proteins are in accord or in conflict with such additional constraints, e.g., amino acid conservation or mutations.

The effect of labelling restrictions strongly depends on how they are distributed in the sequence. If the labelling restrictions are scattered and not localized in extended continuous chunks, then a forbidden labelling position is easily replaced by a neighboring residue, resulting in a negligible loss of information of less than 0.1\AA <<RMSD>>. If the restrictions are grouped together and block a whole “rigid element”, then the uncertainty for that element is fully determined by the computational modelling. We demonstrate this on the example of T4L. The algorithm was allowed to pick only from 46 previously measured pairs, whereas in the ideal in-silico scenario $(N^2-N)/2 = 13041$ pairs would be available, where $N = 162$ is the number of amino acids for T4L. A more systematic investigation of the effect of additional biochemical constraints on the pair selection and uncertainty of the structural models would be a direction for future research.

4. Fig. 7 seems to miss information that is described in the caption. I didn't find the scatter plots in the upper left corner.

We meant <<RMSD>> vs #pairs decay plots. We fixed the caption.

5. Finally a general remark: While structure is important to understand protein function, this understanding often requires knowledge of the precise arrangement of side chains, clearly information that cannot be provided by a low-resolution technique such as FRET. For less detailed problems, e.g., overall domain arrangements or assemblies of complexes, I'd guess that also a less detailed FRET-analysis would be adequate.

Indeed, FRET data does not provide information on the orientation of the side-chains. However, given an accurate backbone conformation ($\sim 2\text{\AA}$), typically, positions of the side-chains can be refined by standard molecular dynamics simulations with time lengths that are modest by current standards ($< 1\ \mu\text{s}$). Additionally, dead end elimination algorithms using secondary structure-specific rotamer libraries allow placing side chains in compact areas very accurately given (good) backbone structures. We now mention the refinement of side-chain orientations by MD or other techniques in the Discussion and Methods sections (pp. 15 and 23).

REVIEWERS' COMMENTS:

Reviewer #1 (Remarks to the Author):

The authors have effectively responded to all the reviewers' comments. The revisions have improved the manuscript. I do not see any issues to prevent publication. This paper is a solid description of a complete workflow to enable single molecule FRET experiments to guide and constrain modeling of biomolecular configurations.